# INCENTIVIZED COLLABORATIVE LEARNING: ARCHITECTURAL DESIGN AND INSIGHTS

## ABSTRACT

Collaborations among various entities, such as companies, research labs, AI agents, and edge devices, have become increasingly crucial for achieving machine learning tasks that cannot be accomplished by a single entity alone. This is likely due to factors such as security constraints, privacy concerns, and limitations in computation resources. As a result, collaborative learning (CL) research has been gaining momentum. However, a significant challenge in practical applications of CL is how to effectively incentivize multiple entities to collaborate before any collaboration occurs. In this study, we propose ICL, an architectural framework for incentivized collaborative learning, and provide insights into the critical issue of when and why incentives can improve collaboration performance. Then, we apply the concepts of ICL to specific use cases in federated learning, assisted learning, and multi-armed bandit, corroborated with both theoretical and experimental results.

## 1 INTRODUCTION

Over the past decade, artificial intelligence (AI) has achieved significant success in engineering and scientific domains, e.g., robotic control (Deisenroth et al., 2013; Kober & Peters, 2014), natural language processing (Li et al., 2016; Bahdanau et al., 2017), computer vision (Liu et al., 2017; Brunner et al., 2017), and finance (Lee et al., 2020; Lussange et al., 2021). With this trend, a growing number of entities, e.g., governments, hospitals, companies, and edge devices, are integrating AI models into their workflows to facilitate data analysis and enhance decision-making.

While a variety of standardized machine learning models are readily available for entities to implement AI tasks, model performance heavily depends on the quality and availability of local training data, models, and computation resources (Goodfellow et al., 2016). For example, a local bank's financial model may be constrained by the small size of its subjects and the number of feature variables. However, this bank could possibly improve its model by integrating additional observations and feature variables from other banks or industry sectors. Therefore, there is a strong need for collaborative learning that allows entities to enhance their model performance while respecting the proprietary nature of local resources. This has motivated recent research on learning frameworks, such as Federated Learning (FL) (Konecny et al., 2016; McMahan et al., 2017; Diao et al., 2021b; 2022c) and Assisted Learning (AL) (Xian et al., 2020b; Chen et al., 2023b; Diao et al., 2022b; Wang et al., 2022), which can improve learning performance from distributed data.

### 1.1 DESIGN GOALS OF INCENTIVIZED COLLABORATIVE LEARNING

Machine learning entities, similar to humans, can collaborate to accomplish tasks that benefit each participant. However, these entities possess local machine learning resources that can be highly heterogeneous in terms of training procedures, computation cost, sample size, and data quality. A key challenge in facilitating such collaborations is understanding the motivations and incentives that drive entities to participate in the first place. For example, a recent study on parallel assisted learning (Wang et al., 2022, Section III.D) has demonstrated cases where two entities, Alice and Bob, collaborate to improve the performance of their distinct tasks simultaneously. However, it may be the case that Alice can assist Bob more than the other way around. In such scenarios, an effective incentive mechanism is crucial for facilitating a "benign" collaboration in which high-quality entities are suitably motivated to maximize the overall benefit.

The need to deploy collaborative learning systems in the real world has motivated recent research in incentivized FL. The existing literature has studied different aspects of incentives in particular

application cases, e.g., using contract theory to set the price for participating in FL, or evaluating contributions for reward or penalty allocation, which we will review in Subsection 1.2. However, understanding when and why incentive mechanism design can enhance collaboration performance is under-explored. This work aims to address this problem by developing an incentivized collaborative learning framework and general principles that can be applied to many use cases. Three examples are given below to show the potential use scenarios to be revisited in later sections.

**Example 1**. Multiple clinics hold data on different patients. They must decide whether to participate in an FL platform to improve their local prediction performance. Specifically, each participating clinic will have the chance to be selected as an active participant to realize an FL-updated model using their local data and computation resources. All participants will receive the updated model. A clinic has the incentive to participate as long as the expected model gain is more valuable than the participation price it must pay for the platform, other participants, or both. From the system perspective, the incentive aims to maximize the model gain or monetary profit from entities' payments.

**Example 2**. Multiple entities collect surveys from the same cohort of customers but in different modalities, e.g., demographic and healthcare information. The user data can be collated by a non-private identifier, e.g., an encrypted username. They will use assisted learning to improve predictability by leveraging multi-modalities without sharing models. The incentive mechanism must enable entities to reach a consensus on who to realize the collaboration to benefit the entities.

**Example 3**. An investment team is recruiting strategists. Once recruited, a strategist will be selected by a multi-armed bandit (MAB) rule to realize a reward that will be enjoyed by all the participants. Each strategist will pay or be paid to participate. Under a good incentive, 1) a top-performing strategist may participate because it may be selected and receive collected payment from others, 2) a mediocre strategist may participate to enjoy the reward shared by the top strategist at a relatively smaller participation cost, and 3) a laggard strategist may not participate because of an overly high participation cost, which can in turn benefit the collaborative system by reducing exploration costs.

This work establishes an incentivized collaborative learning framework to abstract common application scenarios. In our framework, as illustrated in Fig. 1 and elaborated in Section 2, a set of learners play three roles as the game proceeds: 1) *candidates*, who decide whether to participate in the game based on a pricing plan, 2) *participants*, whose final payment, which can be negative if interpreted as a reward, depends on the pricing plan and actual outcomes of collaboration, and 3) *active participants*, who jointly realize a collaboration gain to be enjoyed by all participants.

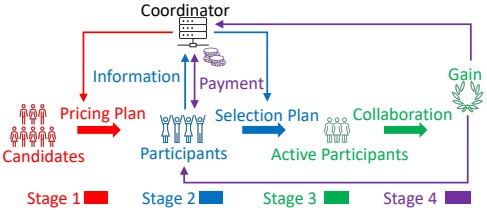

Figure 1: An overview of the incentivized collaborative learning game

The system-level goal is to promote high-quality collaborations for an objective. Examples of the collaboration gain include improved models, predictability, and rewards.

We will use the following design principles of incentives to benefit collaboration: (1) Each participant can simultaneously play the roles of contributor and beneficiary of the collaboration gain. (2) Each participant will pay to participate in return for a collaboration gain if the improvement over its local gain outweighs its participation cost. (3) The pricing plan determines each entity's participation cost and can be positive or non-positive, tailored to reward those who contribute positively and charge those who hinder collaboration or disproportionately benefit from it. (4) The system for collaboration may incur a net zero cost, while still engaging entities to achieve the maximum possible collaborative gains. Our framework provides a unified understanding for the modular design of incentives from a system design perspective. We will show how incentives can be used to reduce exploration complexity and create win-win situations for participants from collaboration.

**Contribution of this work**: The main contributions are threefold.

• First, we propose an architectural framework for incentivized collaborative learning (ICL) along with design principles. These collectively formalize the role of incentives in learning, ensuring that eligible entities are motivated to actively foster collaboration and benefit all the participants.

• Second, we apply the proposed framework and principles to three concrete use scenarios, including incentivized FL, AL, and MAB, and quantify the insights. In particular, we demonstrate how the pricing and selection plans can play crucial roles in reducing the cost of exploration in learning, ul-

timately creating mutual benefits for all participating entities. We also conduct experimental studies to corroborate the developed theory and insights.

• Lastly, the developed ICL framework promotes interoperability among different collaborative learning settings, allowing researchers to adapt the same architecture across various use cases.

## 1.2 RELATED WORK

To address collaborative learning challenges, existing studies have focused on various aspects, such as security (Bhagoji et al., 2019; Bagdasaryan et al., 2020), privacy (Truex et al., 2019; Le et al., 2022), fairness (Yu et al., 2020b;a), personalization (Chen et al., 2022; Le et al., 2022; Khan et al., 2023), model heterogeneity (He et al., 2020; Diao et al., 2021b; Mushtaq et al., 2021), and lack-of-labels (Diao et al., 2022c; Mushtaq et al., 2023). However, a fundamental question remains: why would participants want to join collaborative learning in the first place? This has motivated an active line of research to use incentive schemes to enhance collaboration. We briefly review them below.

**Promoter of incentives** Who want to design mechanisms to incentivize participants and initiate a collaboration? From this angle, existing work can be roughly categorized in two classes: server-centric, meaning that a collaboration is initiated by a server who owns the model and aims to incentivize edge devices to join model training (Kang et al., 2019; Pandey et al., 2019; Zeng et al., 2020; Zhan et al., 2020), and participant-centric where the incentives are designed at the participants' interest (Huang et al., 2022).

**Different goals of incentives** What is the objective of an incentive mechanism design? Most existing work on incentivized collaborative learning, in particular FL, have adopted some common rules for incentive mechanism design, e.g. incentive compatibility and individual rationality (Kang et al., 2019; Zhan et al., 2020). The eventual objective for incentivized collaboration is often maximizing profit from the perspective of the incentive mechanism designer, which is either the coordinator (also called "server", "platform") (Kang et al., 2019) or the participants (also called "clients" in FL) (Huang et al., 2022). Another commonly studied objective is maximizing global model performance in FL, which can be commercialized and turned into profit (Zeng et al., 2020; Yu et al., 2020b; Cho et al., 2022). Other objectives being studied include budget balance (Tang & Wong, 2021; Zhang et al., 2022), computational efficiency (Xu et al., 2015; Zhang et al., 2022), fairness (Tay et al., 2022), and Pareto efficiency (Zeng et al., 2020).

**Components of incentive mechanism design** How to implement incentives? The existing work has focused one of the following machinery: 1) pricing, in which participants bid for collaboration (using the auction theory) (Zeng et al., 2020; Zhang et al., 2022) or a coordinating server determines the price (based on contract theory) (Kang et al., 2019; Ye et al., 2022),and 2) payment allocation, based on contribution evaluation (Yang et al., 2022), fair sharing (Carbunar & Tripunitara, 2010; Yu et al., 2020a), rewarding, or local accuracy (Han et al., 2022).

Overall, the role of incentives in collaborative learning has inspired many recent studies on bringing economic concepts to design learning platforms. Most existing work has focused on FL, especially mobile edge computing scenarios. Nonetheless, the need for collaboration extends beyond FL, as shown in (Tay et al., 2022) which studied synthetic data generation based on collaborative data sharing, and (Wang et al., 2022) which developed an AL framework where an entity being assisted is bound to assist others based on implicit mechanism design. Moreover, incentives in collaborative learning is under-studied in two critical aspects. Firstly, how to design incentives under a unified architecture, considering the existing work often focuses on specific application scenarios? Secondly, when and why do incentives improve collaboration performance? Prior work has often focused on designing an incentive as a separate problem based on an existing collaboration scheme, instead of treating incentive as part of the learning itself. These gaps motivated this work on ICL.

## 2 FORMULATION OF INCENTIVIZED COLLABORATIVE LEARNING (ICL)

### 2.1 OVERVIEW OF ICL AND INTUITIONS

We will focus on a generic round and provide an overview of the ICL formulation below. As illustrated in Fig. 1, a collaboration consists of four stages. In Stage 1, the coordinator sets a pricing plan based on prior knowledge of the candidates' potential gains (e.g., from previous rounds), and each candidate decides whether to be a participant by committing a payment at the end of this round. In Stage 2, the coordinator collects participants' information (e.g., their estimated gains) and uses a selection plan to choose the active participants. In Stage 3, the active participants collaborate to produce an outcome, which is enjoyed by all participants (including non-active ones). In Stage 4,

the coordinator charges according to the pricing plan, the *realized* collaboration gain, and individual gains of active participants. Here, a gain (e.g., decrease in test loss) is assumed to be a function of the realized outcome (e.g., trained model).

## 2.2 DETAILED EXPLANATIONS OF THE ICL COMPONENTS

The proposed collaborative learning game includes two parties: candidate entities and coordinator. For notational convenience, we will first introduce a single-round game and extend it in Section 3.

**Candidates**: Consider $M$ candidates indexed by $[M] \triangleq \{1, \ldots, M\}$. In an ICL game, each candidate $m$ can *potentially* produce an outcome $x_m \in \mathcal{X}$, such as a model parameter. Any element in $\mathcal{X}$ can be mapped to a gain $Z \in \mathbb{R}$, e.g., reduced prediction loss. But such a gain will not necessarily be realized unless the candidate becomes an active collaborator of the game. At the beginning of a game, a candidate will receive a pricing plan from the coordinator specifying the cost of participating in the game and use that to decide whether to become a *participant* of the game. If a candidate participates, it has the opportunity to be selected as an *active participant*. All active participants will then collaborate to produce an outcome (e.g., model or prediction protocol), which also generates a collaboration gain. This outcome is distributed among all participants to benefit them. At the end of the game, all participants must pay according to the pre-specified pricing plan, with the actual price depending on the realized collaboration gain.

We let $\mathbb{I}_P$ and $\mathbb{I}_A$ denote the set of participants and active participants, respectively (so $\mathbb{I}_A \subseteq \mathbb{I}_P \subseteq [M]$). Given the above, an entity has a *consumer-provider bi-profile* in the sense that it can serve as a consumer who wishes to benefit from and also a provider who contributes to the collaboration.

**Coordinator**: A coordinator, e.g., company, government agency, or platform, orchestrates the game by performing the following actions in order: determine a pricing plan of the participation costs based on initial information collected from candidates, select active participants from those candidates that have chosen to become participants, realize the collaboration gain, and charge the participants according to the gain. The coordinator can be a virtual entity rather than a physical one.

**Collaboration gain**: Given active participants represented by $\mathbb{I}_A$, the collaborative gain is a function of their individual outcomes, denoted by $\mathcal{G} : (x_m, m \in \mathbb{I}_A) \mapsto z_{\mathbb{I}_A} \in \mathbb{R}$. This gain will be enjoyed by all participants and the coordinator, e.g., in the form of an improved model distributed by the coordinator. We also use $\mathcal{G} : x_m \mapsto z_m \in \mathbb{R}, m \in [M]$, to denote the gain of an individual outcome.

**Pricing plan**: The pricing plan is a function from $\mathbb{R}^{|\mathbb{I}_A|+1}$ to $\mathbb{R}^{|\mathbb{I}_P|}$ that maps the collaboration gain and individual gains of active participants to a cost needed to participate in the game, denoted by

$$\mathcal{P} : (z, z_m, m \in \mathbb{I}_A) \mapsto (c_j, j \in \mathbb{I}_P), \tag{1}$$

where $z$ denotes the realized collaboration gain. In practice, we may parameterize $\mathcal{P}$ so that it is low for active and good-performing entities, medium for non-active entities, and high for active and laggard/disruptive entities, a point we will demonstrate in the experiments. We assume that the active participants will share their individual gains, namely $z_m, m \in \mathbb{I}_A$, so that all other participants' cost can be evaluated. The $\mathcal{P}$ will provide incentives to each candidate to decide to participate or not. As such, we denote the set of participants by $\mathbb{I}_P = \text{Incent}(\mathcal{P})$.

**Profit**: For each party, the profit will consist of two components: monetary profit from participation fees and gain-converted profit from collaboration gains. More specifically, let $c_m$ denote the final participation cost for entity $m$. Let the Utility-Income (UI) function $z \mapsto \mathcal{U}(z)$ determine the amount of income uniquely associated with any particular gain $Z$. We suppose the UI function is the same for participants and the system. Then, the profit of client $m$ is

$$\text{PROFIT}_m \triangleq \mathbb{1}_{m \in \mathbb{I}_P} \cdot (-c_m + \mathcal{U}(z_{\mathbb{I}_A}) - \mathcal{U}(z_m)), \tag{2}$$

where $z_{\mathbb{I}_A} \triangleq \mathcal{G}(x_m, m \in \mathbb{I}_A)$, and the last term contrasts with its standalone learning. We define the system-level profit as the overall income from participation, $\sum_{m \in \mathbb{I}_P} c_m$, weighted plus the amount converted from collaboration gain, $\mathcal{U}(z_{\mathbb{I}_A})$, namely

$$\text{PROFIT}_{\text{sys}} \triangleq \lambda \sum_{m \in \mathbb{I}_P} c_m + \mathcal{U}(z_{\mathbb{I}_A}), \tag{3}$$

where $\lambda \geq 0$ is a pre-specified control variable that balances the monetary income and collaboration gain. We can regard the system-level profit as the coordinator's profit.

**Remark 1** (Coordinator's profit). One may put additional constraints on the coordinator's monetary income $\sum_{m \in \mathbb{I}_P} c_m$. A particular case is to restrict that $\sum_{m \in \mathbb{I}_P} c_m = 0$, which may be interpreted that the system does not need actual monetary income but rather uses the mechanism for model improvement. This is typical in coordinator-free decentralized learning (to revisit in Section 3.2).

**Selection plan**. The coordinator will select the active participants $\mathbb{I}_A$ from $\mathbb{I}_P$ based on a set of available information, denoted by $\mathcal{I}$. We assume that the $\mathcal{I}$ consists of the coordinator's belief of the distributions of $x_m$ (namely the realizable gain) for each client $m$ in $\mathbb{I}_P$. The selection plan is a function that maps from $\mathcal{I}$ and $\mathbb{I}_P$ to a set $\mathbb{I}_A \subseteq \mathbb{I}_P$, denoted by

$$\mathcal{S} : (\mathcal{I}, \mathbb{I}_P) \mapsto \mathbb{I}_A. \tag{4}$$

This can be a randomized map, e.g., when each participant is selected with a certain probability (Section 2.4.2). In practice, $\mathcal{I}$ may refer to the coordinator's estimates of the underlying distribution of $x_m$, $m \in \mathbb{I}_P$, based on historical performance on the participant side.

**Objective of mechanism design**. Our objective in designing a collaboration mechanism is to maximize the system-level profit under constraints tied to candidates' individual incentives, which will be revisited in Section 2.4. The maximization is over the pricing plan $\mathcal{P}$ and selection plan $\mathcal{S}$. With the earlier discussions, the objective is to solve

$$\text{Objective: } \max_{\mathcal{P}, \mathcal{S}} \mathbb{E}\left\{ \lambda \sum_{m \in \mathbb{I}_P} c_m + \mathcal{U}(z_{\mathbb{I}_A}) \right\}, \text{ where} \tag{5}$$

$$c_m \text{ is specified by } \mathcal{P} \text{ in (1)}, \ z_{\mathbb{I}_A} = \mathcal{G}(x_m, m \in \mathbb{I}_A), \ \mathbb{I}_A = \mathcal{S}(\mathcal{I}, \mathbb{I}_P), \text{ s.t. } \mathbb{I}_P = \text{Incent}(\mathcal{P}). \tag{6}$$

We will elaborate on (6) in Section 2.3.

**Remark 2** (Interpretation of the objective). We discuss three interesting cases of the objective. First, when $\lambda = 1$, the objective is equivalent to maximizing the overall profit. Second, when $\lambda = 0$, the objective is to improve the modeling through a collaboration mechanism. In this case, the system has no interest in the participation income but only provides a platform to incentivize the non-active to pay for the gain obtained by the active. Since the system need not pay for any participants, it is natural to assume the "zero-balance" constraint $\sum_{m \in \mathbb{I}_P} c_m = 0$. Thus, we have

$$\text{Objective: } \max_{\mathcal{P}, \mathcal{S}} \mathbb{E}\{\mathcal{U}(z_{\mathbb{I}_A})\}. \tag{7}$$

Third, as $\lambda \to \infty$, the objective reduces to maximizing the system profit, $\max_{\mathcal{P}, \mathcal{S}} \mathbb{E}\{\sum_{m \in \mathbb{I}_P} c_m\}$. Intuitively, the collaboration gain should still be reasonable to attract sufficiently many participants. Lastly, the following proposition shows that by properly replacing the $\lambda$, the system's objective can be interpreted as an alternative objective that combines the system's and participants' gains.

**Proposition 1.** Let $\lambda' \triangleq \frac{\lambda - 1}{|\mathbb{I}_P| + 1}$. The Objective (5) where $\lambda'$ is replaced with $\lambda$ is equivalent to maximizing the average social welfare defined by $(\text{PROFIT}_{\text{sys}} + \sum_{m \in \mathbb{I}_P} \text{PROFIT}_m)/(|\mathbb{I}_P| + 1)$.

## 2.3 INCENTIVES OF PARTICIPATION

We study the incentives of collaboration from the candidates' perspectives. First, we will elaborate on (6) here. For each candidate, the incentive to become a participant is the larger profit of receiving the collaboration gain compared with realizing a gain on its own. Then, candidate $m$ has the *incentive* to participate in the game if and only if

$$\text{Incent}_m : \ \mathbb{E}\left\{ -c_m + \mathcal{U}(z_{\mathbb{I}_A}) - \mathcal{U}(z_m) \right\} \geq 0, \tag{8}$$

where $z_{\mathbb{I}_A}$ and $c_m$ were introduced in (6). Here, $\mathbb{E}$ denotes the expectation regarding the random quantities, including the active participant set and the realized gains.

**Remark 3** (Inaccurate candidate). A candidate may have its own expectation $\mathbb{E}_m$ in place of $\mathbb{E}$ in (8) when making its decision. In this case, if the candidate is overly confident about the collaboration gain – its expected $z$ tends to be larger than the actual, either intentionally or not – it will participate in the game. The system can have a further screening of it: 1) if this participant is selected as an active participant, it will likely suffer from a penalty since its realized gain will be seen by the coordinator, which will implicitly give feedback as an incentive to that candidate; 2) if not selected, it will become an inactive participant, which will contribute to the system's profit but not harm the collaboration. In this way, a candidate will have a limited extent to harm the system.

## 2.4 Mechanism design for the ICL game

The idea of mechanism design in economic theory is to devise mechanisms to jointly regulate the decisions of multiple parties in a game to eventually attain a system's desired goal (see, e.g., (Maskin, 2008)). In our ICL game, the system's desired goal is to maximize (5), and the mechanisms to design include $\mathcal{P}$ and $\mathcal{S}$. Section 2.3 discussed the incentives from the candidates' view. This subsection studies the mechanism designs from the system's perspective.

### 2.4.1 Pricing plan: from candidates to participants

From the system's view, we can cast the $M$ candidates and the coordinator as the parties in a game. Consider the following strategy choices of each party. Each candidate $m$ has two choices: whether to participate or not, represented by $b_m \triangleq \mathbb{1}_{m \in \mathbb{I}_\mathrm{P}} \in \{0, 1\}$; the coordinator has a choice of the pricing and selection plans, denoted by $(\mathcal{P}, \mathcal{S})$. Following the notation in (4), for a set of participants that exclude $m$, denoted by $\mathbb{I}_\mathrm{P}^{(-m)}$, we let $\mathbb{I}_\mathrm{A}^{(-m)} \triangleq \mathcal{S}(\mathcal{I}, \mathbb{I}_\mathrm{P}^{(-m)})$ and $\mathbb{I}_\mathrm{A} \triangleq \mathcal{S}(\mathcal{I}, \mathbb{I}_\mathrm{P}^{(-m)} \cup \{m\})$. We have the following condition under a Nash equilibrium.

**Theorem 1** (Equilibrium condition). The condition to attain Nash equilibrium is

$$\text{Incent}_m : \ \mathbb{E}\big\{-c_m + \mathcal{U}(z_{\mathbb{I}_\mathrm{A}}) - \mathcal{U}(z_m)\big\} \geq 0, \quad \text{if and only if } m \in \mathbb{I}_\mathrm{P}, \tag{9}$$

$$\text{Incent}_\text{sys} : \ \mathbb{E}\big\{\lambda c_m + \mathcal{U}(z_{\mathbb{I}_\mathrm{A}}) - \mathcal{U}(z_{\mathbb{I}_\mathrm{A}^{(-m)}})\big\} \geq 0, \quad \text{if and only if } m \in \mathbb{I}_\mathrm{P}. \tag{10}$$

**Remark 4** (Pricing as a part of the collaborative learning). A critical reader may wonder why not price participants directly based on the realized gains, which we refer to as post-collaboration pricing, e.g., using the Shapley value (Roth, 1988; Khan et al., 2023). The main distinction is that our studied pricing plan can not only generate profit or reallocate resources on the system side but also influence collaboration gains. Specifically, the pricing plan can screen higher-quality candidates to allow the coordinator to improve model performance in the subsequent collaboration. For instance, consider the case where the sole purpose is to maximize collaboration gain, namely $\lambda = 0$. In this situation, an entity $m$ violating the condition in (10) is treated as a laggard, and $c_m$ can be designed accordingly to ensure this client will not participate, as per violating (9).

### 2.4.2 Selection plan: from participants to active participants

We introduce a general probabilistic selection plan. Assume the information transmitted from participant $m$ is a distribution of $x_m$, denoted by $\mathcal{P}_m$ for all $m \in \mathbb{I}_\mathrm{P}$. Suppose the system expects to select $\rho \in (0, 1]$ proportion of the participants. Consider a probabilistic selection plan that will select each client $m$ in $\mathbb{I}_\mathrm{P}$ with probability $q_m \in [0, 1]$. Let $q = [q_m]_{m \in \mathbb{I}_\mathrm{P}}$. We thus have the constraint

$$q \in Q(\rho, \mathbb{I}_\mathrm{P}) \triangleq \Big\{ q : \sum_{m \in \mathbb{I}_\mathrm{P}} q_m = \rho |\mathbb{I}_\mathrm{P}| \Big\}. \tag{11}$$

Let $b_m$ denote an independent Bernoulli random variable with $\mathbb{P}(b_m = 1) = q_m$, or $b_m \sim \mathrm{B}(q_m)$. Then, conditional on the existing participants $\mathbb{I}_\mathrm{P}$, maximizing any system objective, e.g., (5) and (7), will lead to a particular law of client selection represented by $q$. For example, for the objective (7), we may solve the following problem. $q^* = \arg\max_{q \in Q(\rho, \mathbb{I}_\mathrm{P})} U(q) \triangleq \mathbb{E}\{\mathcal{U}(z_{\mathbb{I}_\mathrm{A}}) = \mathcal{U}(\mathcal{G}(x_m, m \in \mathbb{I}_\mathrm{A}))\}$, where the expectation is over $b_m \sim \mathrm{B}(q_m), x_m \in \mathcal{P}_m$, and $\mathbb{I}_\mathrm{A} = \{m \in \mathbb{I}_\mathrm{P} : b_m = 1\}$. We will show specific examples in Section 3. The existing works have examined client sampling from perspectives other than incentives, such as minimizing gradient variance (Chen et al., 2020).

**Remark 5** (Free-rider and adversarial participants). We will briefly discuss how pricing and selection plans can jointly address concerns regarding free-riders and adversarial participants. A free-rider is an entity $m$ with a low local gain $z_m$ but hopes to participate to enjoy the collaboration gain realized by other more capable participants with a relatively small participation cost. To that end, the entity may deliberately inform the system of a poor $\mathcal{P}_m$ so that the system, if following the above-optimized selection plan, will not select it as active. Consequently, the free-rider's actual local gain will not be revealed and may not suffer a high participation cost. This case motivates us to adopt a random selection to a certain extent in selecting the active participants. More specifically, suppose every participant $m \in \mathbb{I}_\mathrm{P}$ will have at least a $\bar{\rho} > 0$ probability of being selected to be active. Then, it expects to pay at least $\bar{\rho} \cdot \mathbb{E}\{c_m\} = \bar{\rho} \cdot \mathcal{P}(z, z_i, i \in \mathbb{I}_\mathrm{A})$ in return for an additional model gain of $\mathbb{E}\{\mathcal{U}(z_{\mathbb{I}_\mathrm{A}}) - \mathcal{U}(z_m)\}$, where $\mathbb{I}_\mathrm{A}$ contains $m$. Thus, it is not worth the entity $m$'s participation should the system design a cost function that meets the following:

$\bar{\rho} \cdot \mathbb{E}\{\mathcal{P}(z, z_m, m \in \mathbb{I}_A)\} \geq \mathbb{E}\{\mathcal{U}(z_{\mathbb{I}_A}) - \mathcal{U}(z_m)\}$ for all $z_m$ overly small. For example, the coordinator may impose a high cost whenever the realized local gain $z_m$ revealed after the collaboration exceeds a pre-specified threshold. On the other hand, there may be an adversarial participant with a poor local gain but informs the system of an excellent $\mathcal{P}_m$ so that the system will select it to be active. In such cases, the same argument regarding the choice of the pricing plan applies, so no adversarial entity would dare to risk paying an excessively high cost after participating in the game.

# 3 USE CASES OF COLLABORATIVE LEARNING

## 3.1 ICL FOR FEDERATED LEARNING

Federated learning (FL) (Konecny et al., 2016; McMahan et al., 2017; Diao et al., 2021b) is a popular distributed learning framework where the main idea is to learn a joint model using the averaging of locally learned model parameters. Its original goal is to exploit the resources of massive edge devices (also called "clients") to achieve a global objective orchestrated by a central coordinator ("server") in a way that the training data do not need to be transmitted. In line with FL, we suppose that at any particular round, the outcome of client $m$, $x_m$, represents a model. The collaboration will generate an outcome in an additive form: $z_{\mathbb{I}_A} \triangleq \mathcal{G}(x_m, m \in \mathbb{I}_A) = \mathcal{G}(\sum_{i \in \mathbb{I}_A} \zeta_i x_i / \sum_{i \in \mathbb{I}_A} \zeta_i)$, where $\zeta_i$'s are the pre-determined unnormalized positive weights associated with all the candidates, e.g., according to the sample size (McMahan et al., 2017) or uncertainty quantification (Chen et al., 2022). Let $\mathbb{I}_P \triangleq [K]$, where $K \leq M$ is the number of participants, and $M$ is the number of candidates.

### 3.1.1 LARGE-PARTICIPATION APPROXIMATION

Suppose the selection plan is based on a random sampling of $\mathbb{I}_P$ with a given probability, say $\rho \in (0, 1)$, for each participant to be active. Assume $b_m$, for $m \in [K]$, are IID Bernoulli random variables with $\mathbb{P}(b_m = 1) = \rho$. Let $\mathcal{U} \circ \mathcal{G}$ denote the composition of $\mathcal{U}$ and $\mathcal{G}$, and $(\mathcal{U} \circ \mathcal{G})'$ its derivative. Then, we have the following result to approximate the equilibrium conditions in Section 2.4 under a large number of participating clients. Let $\bar{\zeta}_{\mathbb{I}_P}$ and $\bar{x}_{\mathbb{I}_P}$ denote all the participants' average of $\zeta$ and weighted average of $X$, namely $\bar{\zeta}_{\mathbb{I}_P} \triangleq (\sum_{i \in \mathbb{I}_P} \zeta_i)/|\mathbb{I}_P|$, $\bar{x}_{\mathbb{I}_P} \triangleq (\sum_{i \in \mathbb{I}_P} \zeta_i x_i)/(\sum_{i \in \mathbb{I}_P} \zeta_i)$.

**Theorem 2.** Assume $\max\{|\zeta_m|, \|\zeta_m x_m\|_\infty\} \leq \tau$ for all $m \in \mathbb{I}_P$ and $\log d/|\mathbb{I}_P| \to 0$ as $|\mathbb{I}_P| \to \infty$, where $|\mathbb{I}_P|$ is the number of participants. Then, the conditions (9) and (10) can be written as

$$\mathbb{E}\{c_m\} \leq \mathbb{E}\bigg\{ (\mathcal{U} \circ \mathcal{G})(\bar{x}_{\mathbb{I}_P} + o_p(1)) - \mathcal{U} \circ \mathcal{G}(x_m) \bigg\}, \tag{12}$$

$$\lambda \mathbb{E}\{c_m\} \geq \mathbb{E}\bigg\{ (\mathcal{U} \circ \mathcal{G})'(\bar{x}_{\mathbb{I}_P} + o_p(1)) \frac{b_m}{(1 + o_p(1))\rho} \times \frac{\zeta_m}{K \bar{\zeta}_{\mathbb{I}_P}} (\bar{x}_{\mathbb{I}_P} - x_m) \bigg\},$$

where $o_p(1)$ denotes a term whose $\ell_1$-norm converges in probability to zero as $|\mathbb{I}_P|$ goes to infinity.

The proof is based on large-sample analysis, e.g., to show that $(\sum_{i \in \mathbb{I}_P} b_i \zeta_i x_i)/(\sum_{j \in \mathbb{I}_P} b_j \zeta_j)$ is asymptotically close to $\bar{x}_{\mathbb{I}_P}$ for large $|\mathbb{I}_P|$. From the above result and its proof, the system's marginal profit increase attributed to an entity $m$ is approximately

$$b_m \cdot \mathbb{E}\bigg\{ \lambda c_m - (\mathcal{U} \circ \mathcal{G})'(\bar{x}_K) \frac{\zeta_m}{K \bar{\zeta}_K} (\bar{x}_K - x_m) \bigg\}. \tag{13}$$

### 3.1.2 ITERATIVE MECHANISM DESIGN

To optimize the mechanism in practice, the server cannot evaluate the collaboration gain $\mathcal{G}(x_m, m \in \mathbb{I}_A)$ due to its complex dependency on the individual outcomes $x_m$. Alternatively, the server can optimize its mechanism by maximizing the sum of the quantities in (13) as a surrogate of the collaboration gain. The determination of the pricing plan will involve multiple candidates' choices. Since a practical FL system involves multiple rounds, we suppose each client will use the results from the previous rounds to approximate the current strategy set and make the next moves. In Appendix D, we will elaborate on this idea and proposed a concrete incentivized FL algorithm built on Theorem 2.

It is envisioned that a reasonable incentive mechanism can enhance the robustness of FL training. To demonstrate this, we develop an experimental study that involves two types of Byzantine attacks: random modification (Xia et al., 2019) and label flipping (Jebreel et al., 2022). Byzantine client ratios are adjusted to two levels: $\{0.2, 0.3\}$. Among participating clients, the server will select 10% of clients as active clients per round. We studied three incentivized FL (labeled as "ICL") pricing

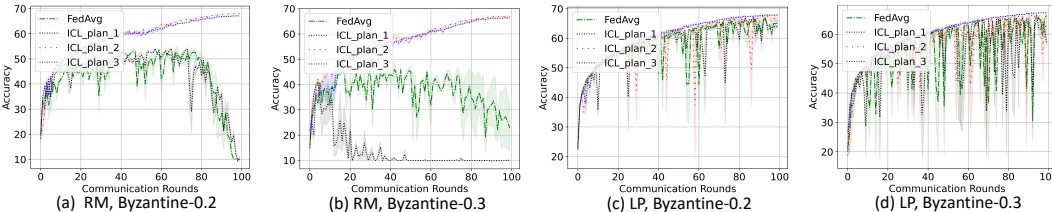

Figure 2: Learning curves of ICL and FedAvg measured by Accuracy, assessed for random modification (RM), label flipping (LP) attacks, and two malicious client ratios (0.2 and 0.3), applied to the CIFAR10 dataset.

plans $\gamma = 1, 2, 3$, where a larger $\gamma$ is interpreted as a higher cost per gain. We visualize the learning curves on the CIFAR10 dataset in Fig. 5. From the results, in comparison with non-incentivized FedAvg, the incentivized FL algorithm from our framework can deliver a higher and more rapidly increasing model performance. On the other hand, ICL with pricing plan 1 underperforms, which is expected as it only imposes a mild penalty on laggard/adversarial active clients. The results also suggest the importance of a reasonable pricing plan. Furthermore, the random modification attack poses a more significant threat compared to the label flipping attack, making it particularly difficult for non-incentivized FedAvg to converge. More extensive results are included in Appendix D.

**Remark 6** (Random sampling (RS) for noninformative scenarios). We show that if the system is noninformative, RS can be close to the optimal selection, following Section 2.4.2. Suppose the information from participant $m$ is the mean and variance of $x_m \in \mathbb{R}$, denoted by $\mu_m, \sigma^2$ for $m \in \mathbb{I}_{\mathrm{P}}$. A large $\sigma^2$ means less information. The result below shows RS is close to the optimal for large $\sigma$.

**Proposition 2.** Assume the gain is defined by $\mathcal{G}(x) \triangleq -\mathbb{E}(x - \mu)^2$, where $\mu$ represents the underlying parameter of interest, and the participants' weights $\zeta_m$'s are the same. Assume that $\sigma^2/(|\mathbb{I}_{\mathrm{P}}| \cdot \max_{m \in \mathbb{I}_{\mathrm{P}}}(\mu_m - \mu)^2) \to \infty$ as $|\mathbb{I}_{\mathrm{P}}| \to \infty$. Then, we have $U(q)/U(q^*) \to_p 1$ as $|\mathbb{I}_{\mathrm{P}}| \to \infty$.

### 3.2 ICL FOR ASSISTED LEARNING

Assisted learning (AL) (Xian et al., 2020b; Diao et al., 2022b; Wang et al., 2022) is a decentralized learning framework that allows organizations to autonomously improve their learning quality within only a few assistance rounds and without sharing local models, objectives, or data. Unlike FL schemes, AL is 1) decentralized in that there is no global model to be shared or synchronized among all the entities in the training and 2) assistance-persistent in the sense that an entity still needs the output from other entities in the prediction stage. From the perspective of incentivized collaboration, the above naturally leads to two considerations with complementary insights into the pricing and selection plans compared with FL in Section 3.1.

• *Consideration 1: Autonomous incentive design without a coordinator* Since each entity can initiate and terminate assistance, it is natural to consider a coordinator-free scenario, where entities can autonomously reach a consensus on collaboration partners based on their pricing plans.

• *Consideration 2: Limited information for incentive design* In AL, an entity aims to seek assistance to enhance prediction performance without sharing proprietary local models. Thus, we suppose the communicated information for collaboration is limited to gains ($z$) rather than outcomes ($x$).

To put it into perspective, we will study a three-entity setting in Section 3.2.1 to develop insights into the incentive that allows for a consensus on collaboration in Stage 1 (Fig. 1). In Section 3.2.2, we will further study a multi-entity setting where multiple less-competitive participants are allowed to enter Stage 2 to enjoy the collaboration gain, but they will not compete for being active participants.

#### 3.2.1 CONSENSUS OF COMPETING CANDIDATES

In this section, we study three candidate entities, Alice, Bob, and Carol, and suppose each candidate aims to maximize its profit. Suppose a collaboration round can only consist of two entities. Then, the collaboration will only occur when two out of the three, say Alice and Bob, can maximally assist each other. From Alice's perspective, Carol is less competitive than Bob, and meanwhile, from Bob's perspective, Carol is less competitive than Alice. We will provide conditions to reach a consensus. Suppose each entity has its own payment plan: entity $i$ will pay a price $p_i(z - z_i)$ for any given collaboration gain $z$ and its local gain $z_i$, for all $i \in [M]$. So, if entities $i$ and $j$ collaborate, the actual price $i$ will pay is $p_i(z - z_i) - p_j(z - z_j)$. The goal of each entity is to maximize the expected gain-converted profit minus the participation cost, namely the quantity in (2). For simplicity, suppose $p_i(\Delta z) = c_i \cdot \Delta z$ and $\mathcal{U}(z) = u \cdot z$. Let $\mu_{i,j} \triangleq \mathbb{E}(\mathcal{U}(z_{i,j}))$ denote the expected income of the collaboration gain formed by entities $i$ and $j$, and $\mu_{j \leftarrow i} \triangleq \mathbb{E}(\mathcal{U}(z_{i,j}) - \mathcal{U}(Z_j))$ the additional gain brought by $i$ to $j$. With this setup, we have the following result.

**Theorem 3.** A consensus on collaboration exists if and only if there exist entities 1 and 2 satisfying

$$(u - c_1)\mu_{1,2} + c_2\mu_{2\leftarrow 1} \geq (u - c_1)\mu_{1,3} + c_3\mu_{3\leftarrow 1}, \tag{14}$$
$$(u - c_2)\mu_{2,1} + c_1\mu_{1\leftarrow 2} \geq (u - c_2)\mu_{2,3} + c_3\mu_{3\leftarrow 2}.$$

Inequality (14) can be interpreted that the total gain of entity 1 received from entity 2, which consists of the collaboration-generated gain $(u - c_1)\mu_{1,2}$ and the pricing-based gain $c_2\mu_{2\leftarrow 1}$, is no larger than that from entity 3. To show our pricing plan can promote mutually beneficial collaboration, we apply ICL to the parallel assisted learning (PAL) framework (Wang et al., 2022) to develop an incentivized PAL. The detailed algorithms and experimental studies are included in Appendix E. We show an experiment using real-world clinical dataset MIMIC (Johnson et al., 2016). Suppose three divisions, denoted by Entity 1, 2, and 3, collect heterogeneous features from the same patients for different tasks: predict the heart rate, the systolic blood pressure, and the length of stay. In Setting $i$, Entity $i$ does not provide an incentive while the other two entities do. The results in Fig. 3 show that in Setting 1, entity 1 will not gain much as it does not provide incentives; in other two settings, it gains from collaborating with the entity with mutual benefits.

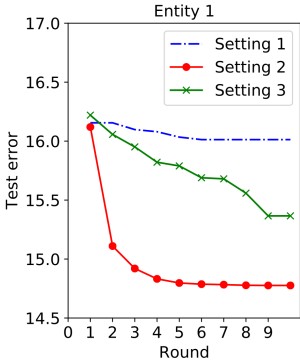

Figure 3: Test error of Entity 1 in three settings. In Setting $i$, entity $i$ pays none ($c_i = 0$) while entity $j \neq i$ pays with $c_j \neq 0$, for $i, j \in [3]$.

### 3.2.2 CONSENSUS OF NON-COMPETING CANDIDATES

We will further study a multi-entity setting where multiple less-competitive participants are allowed to enjoy the collaboration gain, but they will not compete for being active participants. The objective is to develop an incentive to maximize the collaboration gain, namely Objective (7), that will eventually benefit all the participants. For ease of presentation, we will consider only one active participant (namely $|\mathbb{I}_A| = 1$). In general, we may regard a set of participants as one "mega" participant. Following the above two considerations of AL, we will first study the following setup. Suppose $K$ entities decide to participate in a collaboration, where one of them will be selected to realize the collaboration gain. For example, if participant $m$ is active, it will realize a model gain $z_m \sim \mathcal{G}(x_m)$, where $x_m \sim \mathcal{P}_m$ is the potential outcome of participant $m$. Let $\mathcal{P}_m^*$ denote the distribution of $z_m$ induced by $\mathcal{P}_m$ for $m \in [K]$ and suppose they are the *shared* information among participants. In line with Consideration 1, the system is set to have a zero balance, namely $\sum_{m \in [K]} c_m = 0$. Following the notation in (1), we consider the following particular pricing plan:

$$\mathcal{P} : z \mapsto C(z)\mathbb{1}_{j \notin \mathbb{I}_A} - (K - 1)C(z)\mathbb{1}_{j \in \mathbb{I}_A}, \ j \in \mathbb{I}_P, \tag{15}$$

where $C$ is non-decreasing so that a larger gain is associated with a larger cost. In other words, each of the $K - 1$ non-active participants will pay a cost of $C(z)$, which depends on the realized $z$, to the active participant. Then, the condition to reach a consensus among participants is the existence of a participant, say participant 1, such that when it is active, (i) the collaboration gain is maximized, and (ii) every participant sees that its individual profit is maximized, as formalized below.

**Theorem 4.** Assume $\mathcal{U}$ is a pre-specified non-decreasing function. Consider the pricing plan in (15) where $C$ can be any non-negative and non-decreasing function. Let $\mu_m \overset{\Delta}{=} \mathbb{E}_{\mathcal{P}_m^*}\{\mathcal{U}(z_m)\}$ denote the expected gain of participant $m$ when it is active, $m \in [K]$. The necessary and sufficient condition for reaching a pricing consensus is the existence of a participant, say participant 1, that satisfies $\mu_1 - u_j \geq \mathbb{E}_{\mathcal{P}_1^*}\{C(z_1)\} + \mathbb{E}_{\mathcal{P}_j^*}\{(K - 1)C(z_j)\}$ for all $j \neq 1$. If we further assume the linearity $\mathcal{U}(z) \overset{\Delta}{=} u \cdot z$ and $C(z) = c \cdot z$, this inequality becomes $c \leq \min_{j \neq 1} u \cdot (\mu_1 - \mu_j)/(\mu_1 + (K - 1)\mu_j)$.

## 4 CONCLUSION

As the demand for learning tasks continues to grow, collaboration among entities becomes increasingly important for enhancing their performance. We proposed an ICL framework to study how entities can be properly incentivized to create common benefits. While our work provides an angle toward incentivizing collaborative learning, there are several limitations worth further investigation. For instance, future studies could examine the functional forms of pricing plans, use cases to promote model security (Wang et al., 2021; Xian et al., 2023) and privacy (Dwork et al., 2006; Ding & Ding, 2022), and potential trade-offs between collaboration and competition in collaborative learning contexts. The **Appendix** contains more details on ICL and related works, ethics discussion, concrete use cases and experiments of FL, AL, and MAB, and technical proofs.

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

# Appendix for "Incentivized Collaborative Learning: Architectural Design and Insights"

The **Appendix** contains additional discussions on the ICL framework and related work, ethical considerations, detailed use cases of FL, AL, and collaborative MAB, their experimental studies, and all the technical proofs.

## A  FURTHER REMARKS ON THE PROPOSED FRAMEWORK

Next, we will discuss more detailed aspects of the proposed framework.

**Motivation to have a unified framework**. A general incentive framework is important due to the following reasons:

1. Generalizability: A unified framework captures the essence of incentives in various collaborative learning settings, enabling its application across different scenarios. This allows researchers and practitioners to apply the same principles and techniques to multiple use cases, saving time and resources. For instance, we applied the ICL framework to federated learning, assisted learning, and an additional case study called incentivized multi-armed bandit, which we introduce in Appendix F. In these cases, we demonstrated how the crucial components of the pricing plan and selection plan could play key roles in reducing exploration complexity from the learning perspective, thereby creating win-win situations for all participants in the collaborative system.

2. Interoperability: A unified incentive framework promotes interoperability among different collaborative learning settings. Researchers and practitioners can build upon the same principles, avoiding the need to reinvent the wheel for every new use case.

3. Ease of adaptation: A unified framework provides a structured approach to design incentives under a general collaborative learning framework with guaranteed incentive properties. This makes adapting the framework for specific use cases easier, such as defining collaboration gain based on model fairness if that is part of the system-level goal.

In summary, a unified incentive framework for collaborative learning offers generalizability, interoperability, and ease of adaptation. These advantages make it a more appealing approach compared to designing specifically tailored incentive mechanisms for each use case.

**Lie about the realized gain**. Participants may lie about the realized gain to pay less than they have to. This is an important aspect to consider in practical collaborative systems. There are several possible ways to encourage honest behavior in collaborative learning. First, if the realized gain must be published at the end of the collaboration (e.g., due to third-party verification, or the final collaboration gain is a function of individual realized gains), we can design an incentive mechanism to discourage dishonest behavior by rewarding clients who turn out to be honest at the end of the learning process. This type of expected reward can motivate clients to behave well. For example, when defining the "Collaboration gain" in Section 2.2, we considered a generic scenario that the final collaboration gain depends on the transparent outcomes of active participants, which indicates that the system has a way to evaluate the realized gains. Additionally, our incentive framework allows the active participants to gain from non-active participants' paid costs. In that particular case, the active participants' realized outcomes are transparent to the coordinator, so they cannot lie, while the non-active participants do not realize anything but only enjoy the collaboration gain, so they may have to pay a flat rate regardless of whether they lie.

There are several ways to verify clients' realized gain in FL settings, especially FL with heterogeneous clients (e.g., (Diao et al., 2021a) and the references therein). One way is to estimate the realized local gain from global/transparent testing data; Another possible way is to estimate the realized gain from local test data which is uploaded to a verified third party before training. In our work, in the FL setting, the system knows the submitted local models and can estimate their realized gains.

**Heterogeneous contributions to entities**. The contribution to one entity might not be the same as the contribution in another. In a setting where participants have heterogeneous evaluation metrics

(e.g., due to distributional heterogeneity or personalization need), one can let the utility-income function depend on specific entities. For example, in Eq. (9) of the main paper, we can add a subscript of "m" to the $U$-function to highlight its possible dependence on entities. Such a change will be notational, as it will not affect the key ideas and insights in the proposed work. For instance, the idea of using incentives to regulate entities' local decisions still applies.

**Sub-selling of models**. There may be a participant in one round that sub-sell the trained model. This concern can be addressed in three aspects. First, when the utility-income function is defined (by domain experts), it can already account for all the values of a trained model, including its potential subselling. Second, although participants may use a global model to generate synthetic data or apply knowledge distillation for potential performance improvement, these approaches may require spend extra costs in data and modeling, which may not be worth it for the client. Third, even if a participant can leverage an earlier round to better position itself in the next round, we are fine with it as long as it benefits the system – we allow each client to have varying capability in each round.

**Collaboration among competitors**. In practice, there is a possible revenue loss due to competitors getting better. Modeling competing entities in a collaborative learning framework is an interesting future problem. A recent work on MarS-FL framework (Wu & Yu, 2022) introduced $\delta$-stable market and friendliness concepts to assess FL viability and market acceptability. In our framework, one possible formulation of the potential collaborators have conflicting interests is to define the collaboration gain as a function of not only active participants' outcomes but also their level of conflicts.

**Explicit coordinator in the incentivized learning**. We do not need an explicit coordinator in the incentivized learning We employ the term "coordinator" to represent the role of the collaboration initiator or promoter, which could either be a physical coordinator (e.g., the server in an FL setting) or one of the participants (e.g., the entity initiating collaboration in an AL setting). Therefore, we do not really need a physical coordinator in the incentivized learning for all the models. Also, the developed concepts (e.g., entities' incentives, pricing mechanism, selection mechanism) and the derived insights do not depend on the coordinator.

**Timing for each participant to pay**. The general idea of mechanism design is to develop incentives to regulate entities' local decisions to better achieve a global objective. In our framework, a critical component of mechanism design is to use the pricing plan before collaborative learning to provide a suitable incentive for improving collaboration. For transparency, we assume the pricing plan can inform the participants of what they actually would be paying. The pricing plan (mapping from outcome to price) is determined before each candidate entity decides participation or not, as shown in Fig. 1. Additionally, since every entity's outcome could be random, their decisions are all based on expected values.

**Determination of the value of machine learning results**. Assessing the relationship between utility and income in practical applications can be challenging. This is analogous to defining "utility" in economic studies, as its precise value depends on the particular context. For instance, the monetary value of a machine learning model can vary depending on its commercial viability.

## B    ETHICS DISCUSSION

A critical reader may raise the fairness concern that the studied incentive design be unfair and potentially worsen the inequality between high-quality and low-quality entities. We would like to respectfully point out that this has no conflict with the scope of our work.

Firstly, the literature contains numerous definitions of fairness (Shi et al., 2023). It is impractical to evaluate the fairness of a procedure without identifying a specific type of fairness and examining it within the context of a particular use case. We do not see that the proposed framework conflicts with some popular types of fairness. One common type of fairness is model fairness, meaning that the model performs equally well on different subpopulations. In the incentivized collaborative learning framework, there are multiple ways to incorporate such fairness when defining model performance (e.g., using weighted accuracy across different subpopulations). Another type of fairness, more widely studied in game-theoretic settings (Rabin, 1993; Zheng & Peng, 2005) and recently in federated learning contexts (Lyu et al., 2020), is collaboration fairness. Rooted in the game theory (Rabin, 1993), this fairness notion dictates that a client's payoff should be proportional to its contribution to the FL model. Hence, a system that rewards higher-quality contributors more is in-

herently "fair." This direction of fairness aligns with our framework since the goal of the incentive here is to encourage high-quality clients to actively contribute to the collaboration.

Secondly, even though our incentive mechanism encourages high-quality entities to actively contribute, it does not mean that low-quality entities will be "unfairly" treated. In fact, our framework advocates for a win-win situation in which all participants can enjoy the collaboration's results. Keep in mind that each entity's final profit/gain consists of the part attributed to the participation cost/reward and the part attributed to the collaboration gain. If the latter is higher due to a more reasonable incentive, everyone can enjoy positive post-collaboration gains. Therefore, a performance-oriented system goal does not necessarily violate fairness from the perspective of individual participants.

Thirdly, several published works have used the contribution to the model to allocate rewards (e.g., (Yang et al., 2022) and references therein). Related to that direction of work, we aim to design the mechanism of allocation before collaborative learning has trained a model, to provide a suitable incentive for improving the modeling. The collaborative result would then benefit every participant.

In summary, concerning fairness, our work aligns more with contribution fairness (also called collaboration fairness in FL (Zheng & Peng, 2005)). Our motivation for incentives is to suitably regulate the entities' local decisions (who will autonomously make decisions) to facilitate collaborative performance that will be enjoyed by even low-quality entities (but willing to pay), thus creating win-win collaborations. The performance can be measured in various ways depending on practical considerations, such as reflecting model fairness when defining the quality of collaboration, but this is beyond the scope of this work.

## C    MORE DISCUSSIONS ON RELATED WORKS

Table 1: Summary of the most related literature on incentive-based collaboration.

| Setting | Incentive | Key feature |
| --- | --- | --- |
| Federated learning | FLI (Yu et al., 2020a) | fair reward distribution |
| Federated learning | FMore (Zeng et al., 2020) | multi-factor utility function |
| Federated learning | CrossSiloFLI(Tang & Wong, 2021) | social welfare maximization |
| Federated learning | FedAB (Wu et al., 2023) | multi-armed bandit for client selection |
| Assisted learning | AL (Xian et al., 2020a), GAL (Diao et al., 2022a) | intertwined interest in prediction |
| Assisted learning | PAL (Wang et al., 2022) | intertwined interest in training and prediction |
| Collaborative generative modeling | CGM (Tay et al., 2022) | synthetic data as rewards |
| Crowdsourcing | CML (Sim et al., 2020) | model as rewards |
| Crowdsourcing | MSensing (Yang et al., 2012) | reverse auction-based mechanism |
| Decentralized learning | iDML (Yu et al., 2023) | blockchain-based mechanism |

Our proposed ICL framework builds on the related literature on incentivized collaborative systems, which often focuses on one or more of three components: pricing, payment allocation, and participant selection (Zeng et al., 2021). Many incentive mechanisms have been designed for these components, depending on particular application scenarios. In Table 1, we summarize a few representative works and their main features. Next, we will relate these existing works to our proposed ICL framework.

In (Yu et al., 2020a), a fairness-aware incentive scheme for FL called FLI was established to maximize collaboration gain while minimizing inequality in payoffs. The payoff allocation in FLI can be regarded as a particular pricing plan designed to promote both FL model gain and equitable reward allocation. In (Zeng et al., 2020), the authors proposed an FL incentive scheme called FMore that utilizes a multi-dimensional auction to encourage participation from high-quality edge nodes. The scheme determines a utility function based on multiple client-specific factors, which provides a general approach to designing our pricing plan. In (Tang & Wong, 2021; Blum et al., 2021), the authors formulate a particular social welfare objective problem under a budget balance constraint to address the challenges of organization heterogeneity in cross-silo FL. Recently, (Wu et al., 2023) developed an auction-based combinatorial multi-armed bandit algorithm called FedAB for FL client selection to maximize the server's utility. In our framework, this approach can be particularly helpful when the server lacks trust in the clients' shared information. In (Xian et al., 2020a; Diao et al., 2022a), the assisted entity continually relies on others' provided information during the prediction stage, leading

to persistent collaboration even after training. The work of (Wang et al., 2022) develops an assisted learning approach for entities with different learning objectives/tasks, requiring them to combine their tasks during the training phase and jointly decode the prediction results during the prediction stage. As a result, the approach enforces an implicit mechanism design that compels an assisted entity to assist others. In (Tay et al., 2022), the authors proposed a collaborative learning problem to incentivize data contribution from entities for training generative models. The synthetic data is treated as rewards in our framework to solicit contributions from private data sources, different from the commonly used trained models or monetary rewards. In (Sim et al., 2020), the collaboration system uses the Shapley value and and information gain to assess the post-collaboration contribution of participants and charge accordingly. Two particular incentive models for mobile phone sensing systems are developed in (Yang et al., 2012) to encourage user participation. The platform-centric approach uses a Stackelberg game-based mechanism to approximate a solution to Nash equilibrium in a generic round. The user-centric approach employs an auction-based mechanism to determine entity participation. In (Yu et al., 2023), the authors proposed a decentralized and transparent learning system resistant to fraud or manipulation by utilizing blockchain technology. This approach can be compatible with our ICL framework and can be used for secure information sharing. We refer to (Zeng et al., 2021; Tu et al., 2022; Németh et al., 2022; Witt et al., 2022) for more literature reviews.

## D  ICL FOR FEDERATED LEARNING

**Federated Learning (FL):** For an entity to achieve near-oracle performance as if all the local resources were centralized, many distributed learning methods have been developed. A popular direction of research is FL (Konecny et al., 2016; McMahan et al., 2017; Diao et al., 2021b; Ding et al., 2022). The main concept behind FL involves learning a joint model by alternating between the following steps in each federated or communication round: 1) a server sends a model to clients, who subsequently perform multiple local updates, and 2) the server aggregates models from a subset of clients. This approach leverages the resources of a large number of edge devices, also known as "clients," to achieve a global objective coordinated by a central server.

In this section, we give a more specific incentivized FL setup and operational algorithm based on the development in Section 3.1.

### D.1  OPERATIONAL ALGORITHM OF INCENTIVIZED FL

First, let us recall the general Objective (6):

$$\text{Objective:} \quad \max_{\mathcal{P}, \mathcal{S}} \mathbb{E}\left\{ \lambda \sum_{m \in \mathbb{I}_{\text{P}}} c_m + \mathcal{U}(z_{\mathbb{I}_{\text{A}}}) \right\}, \text{ where}$$

$$c_m \text{ was introduced in (1)}, \tag{16}$$

$$z_{\mathbb{I}_{\text{A}}} \triangleq \mathcal{G}(x_m, m \in \mathbb{I}_{\text{A}}), \quad \mathbb{I}_{\text{A}} \triangleq \mathcal{S}(\mathcal{I}, \mathbb{I}_{\text{P}}),$$
$$\text{s.t. } \mathbb{I}_{\text{P}} = \text{Incent}_m(\mathcal{P}). \tag{17}$$

The above could be regarded as a collaboration game in any particular round of FL. As FL consists of multiple rounds, it is natural to use the previous rounds as a basis for each candidate client to decide whether to participate in the current round. We will use a subscript $t$ to highlight the dependence of a quantity on the FL round. For example, the outcome $x_m$ will be replaced with $x_{m,t}$ for round $t$. More specifically, we consider the following FL equipped with an incentive setup:

• Let the outcome $x_{m,t}$ represent the updated model of participant $m$ at round $t$.

• The "gain" function $\mathcal{G}$ maps a model to the decreased test loss compared with the previous round, so the larger, the better. We assume the FL server (coordinator) holds a test dataset to compute the gain.

• The pricing plan $\mathcal{P}_\theta : (z, z_m, m \in \mathbb{I}_{\text{A}}) \mapsto (c_j, j \in \mathbb{I}_{\text{P}})$ can be written in the following form parameterized by $\theta = [\theta_1, \theta_2]$:

$$c_j \triangleq A_1(z; \theta_1) + \mathbb{1}_{j \in \mathbb{I}_{\text{A}}} A_2(z - z_m; \theta_2), \tag{18}$$

where $A_1, A_2$ are pre-specified functions. The *intuitions* is described as follows. On the one hand, if a participant is non-active, the server will never observe its outcome and local gain. So, it is

reasonable to let all non-active participants pay the same price $A_1(z; \theta_1)$ that depends on the realized collaboration gain. On the other hand, if a participant is active, the server will observe its local gain and thus leverage that information to correct the price by $A_2(z - z_m; \theta_2)$, a function of the improved gain due to collaboration (which can be negative). With such a design, it is possible to incentivize a laggard or adversarial client not to participate–since, otherwise, it will possibly be active and suffer from a large (penalty) cost.

• To optimize the pricing plan parameterized by $\theta$, we need to establish an objective function of $\theta$. Based on the proof of Theorem 2, the marginal gain brought by any client $m$ would be

$$\mathbb{E}\left\{\lambda c_m - (\mathcal{U} \circ \mathcal{G})'(\bar{x}_K) \frac{\zeta_m}{K \bar{\zeta}_K} (\bar{x}_K - x_m)\right\} \tag{19}$$

given that the client $m$ participates in the collaboration. The server would foresee the client's strategy based on Inequality (12): namely, it would participate if

$$\mathbb{E}\{\mathcal{U} \circ \mathcal{G}(\bar{x}_K) - \mathcal{U} \circ \mathcal{G}(x_m) - c_m\} \geq 0.$$

We let $b_m \in \{0, 1\}$ denote the indicator variable of the above condition. Given the above, we propose to optimize $\theta$ that maximizes the total marginal gains over those who will participate:

$$O(\theta) \triangleq b_m \cdot \mathbb{E}\left\{\lambda c_m - (\mathcal{U} \circ \mathcal{G})'(\bar{x}_K) \frac{\zeta_m}{K \bar{\zeta}_K} (\bar{x}_K - x_m)\right\}. \tag{20}$$

However, $b_m$ is not differentiable in $\theta$. For gradient descent-based optimization of $\theta$, we use the sigmoid function with scaling factor $s$, denoted by $\sigma_s(b) = (1 + e^{-b/s})^{-1}$ to approximate it. In other words, we let

$$b_m = \sigma_s\left(\mathbb{E}\{\mathcal{U}(\bar{z}_K) - \mathcal{U}(z_m) - c_m\}\right). \tag{21}$$

The hyperparameter $s$ controls the softness of the approximation, which is tuned in the experiment. Moreover, the collaboration or client-specific gains are not observable at the current round $t$ since the outcomes have not been realized. To address that, we will use the performance from the previous rounds to approximate them. More specifically, we will approximate $\bar{z}_K$ with the server's model gain from the last round, denoted by $\bar{z}_{t-1}$; we approximate $z_m$ with the client's model gain from the last round when it was active, denoted by $\bar{z}_{\tau_{m,t}}$. Here and afterward, $\tau_{m,t} \leq t$ denotes the last time client $m$ was active by time $t$.

## D.2 PSEUDOCODE

The pseudocode is summarized in Algorithm 1.

## D.3 EXPERIMENTAL DETAILS FOR INCENTIVIZED FL

In the experimental studies, we considered the following setup.

**Data** We evaluate the FashionMNIST (Xiao et al., 2017), CIFAR10 (Netzer et al., 2011), and CINIC10 (Darlow et al., 2018) datasets. In Table 2, we present the key statistics of each dataset.

Table 2: Statistics of datasets in our experiments

| Datasets | FashionMNIST | CIFAR10 | CINIC10 |
|---|---|---|---|
| Training instances | 60,000 | 50,000 | 20,000 |
| Test instances | 10,000 | 10,000 | 10,000 |
| Features | 784 | 1,024 | 1,024 |
| Classes | 10 | 10 | 10 |

**Settings** Standard FL can be vulnerable to poor-quality or adversarial clients. It is envisioned that a reasonable incentive mechanism can enhance the robustness of FL training against participation of

---

**Algorithm 1** Incentivized Federated Learning

---

**Input:** Datasets $D_{1:M}$ distributed on $M$ local clients, active rate $\rho \in (0,1]$, number of communication rounds $T$, objective parameter $\lambda$, clients' unnormalized weights $\zeta_{1:M}$, test loss $L$, pricing plan $\mathcal{P}_\theta : (z, z_m, m \in \mathbb{I}_A) \mapsto (c_j, j \in \mathbb{I}_P)$, where $\theta = [\theta_1, \theta_2]$ is the unknown parameter, $c_j \stackrel{\Delta}{=} A_1(z; \theta_1) + \mathbb{1}_{j \in \mathbb{I}_A} A_2(z - z_m; \theta_2)$, and $A_1, A_2$ are pre-specified functions, server's test dataset $D_{\text{test}}$, sigmoid function $\sigma_s : v \mapsto (1 + e^{-v/s})^{-1}$ where $s > 0$ is a hyperparameter, learning rate $\eta > 0$ for optimizing $\theta$. Recall that $\tau_{m,t} \leq t$ deni £ select it as active©ˆ§´1············7· ··‘·· · 1· szeeszzsssssssssssssslast round when the client $m$ was active before round $t$, and $\mathcal{U}$ is a given Utility-Income function known to all the clients and the server.

**Output:** Server's updated model $\bar{x}_t$, overall realized profit $\lambda \sum_{m \in \mathbb{I}_{P,t}} c^*_{m,t} + \mathcal{U}(\bar{x}_t)$, $t = 1, \ldots, T$

**Initialization:** $\theta_0, \bar{x}_0, \bar{z}_0 \stackrel{\Delta}{=} \mathcal{G}(\bar{x}_0)$

**System executes:**

  **for** each communication round $t = 1, \ldots, T$ **do**

  $\quad \theta_t \stackrel{\Delta}{=} (\theta_{1,t}, \theta_{2,t}) \leftarrow$ **ServerStrategy**$(\bar{z}_{\tau_{m,t}}, z_{m,\tau_{m,t}}, \bar{x}_{\tau_{m,t}}, x_{\tau_{m,t}}, \zeta_m, m \in [M], \theta_{t-1})$

  $\quad$ Distribute $\mathcal{P}_t \equiv \mathcal{P}_{\theta_t}$ to all $M$ clients

  $\quad b_{m,t} \leftarrow$ **ClientStrategy**$(\mathcal{P}_t, z_{m,\tau_{m,t}}, \bar{z}_{t-1}), m \in [M]$

  $\quad \mathbb{I}_{P,t} \leftarrow \{m \in [M] : b_{m,t} = 1\}$

  $\quad \mathbb{I}_{A,t} \leftarrow$ randomly sample $\max(\lfloor \rho \cdot |\mathbb{I}_{P,t}| \rfloor, 1)$ active clients from $\mathbb{I}_{P,t}$

  $\quad$ **for** each client $m \in \mathbb{I}_{A,t}$ in parallel **do**

  $\quad\quad$ Distribute the server's model parameter $\bar{x}_{t-1}$ to local client $m$

  $\quad\quad x_{m,t} \leftarrow \text{ClientUpdate}(D_m, \bar{x}_{t-1})$ // use any standard local update

  $\quad\quad$ Send $x_{m,t}$ to the server

  $\quad$ **end**

  $\quad$ Receive model parameters from active clients, and calculate $\bar{x}_t = (\sum_{m \in \mathbb{I}_{A,t}} \zeta_m)^{-1} \sum_{m \in \mathbb{I}_{A,t}} \zeta_m x_{m,t}$

  $\quad$ Calculate the collaboration gain $\bar{z}_t = \mathcal{G}(x_{m,t}, m \in \mathbb{I}_A)$ and broadcast to all candidate clients

  $\quad$ Calculate the individual gain of each active participant $z_{m,t} = \mathcal{G}(x_{m,t})$ and return it to the associated client $m$

  $\quad$ Participant $m$ pays $c^*_{m,t} \stackrel{\Delta}{=} A_1(\bar{z}_t; \theta_{1,t}) + \mathbb{1}_{m \in \mathbb{I}_{A,t}} \cdot A_2(\bar{z}_t - z_{m,t}; \theta_{2,t})$, for all $m \in \mathbb{I}_{P,t}$

  **end**

**ServerStrategy** $(\bar{z}_{\tau_{m,t}}, z_{m,\tau_{m,t}}, \bar{x}_{\tau_{m,t}}, x_{\tau_{m,t}}, \zeta_m, m \in [M], \theta_{t-1})$**:**

  Define the objective function of $\theta$ to maximize:

$$O_t(\theta) = \sum_{m=1}^M \sigma_s(\delta_{m,t}) \cdot \left\{ \lambda \cdot c_{m,t}(\theta) - \frac{\zeta_m}{\sum_{j \in \mathbb{I}_{A,\tau_{m,t}}} \zeta_j} \cdot f'(\bar{x}_{\tau_{m,t}}) \cdot (\bar{x}_{\tau_{m,t}} - x_{m,\tau_{m,t}}) \right\}, \quad \text{where}$$

$$c_{m,t}(\theta) \stackrel{\Delta}{=} A_1(\bar{z}_{\tau_{m,t}}; \theta_1) + \rho \cdot A_2(\bar{z}_{\tau_{m,t}} - z_{m,\tau_{m,t}}; \theta_2) \tag{22}$$

$$\delta_{m,t} \stackrel{\Delta}{=} \mathcal{U}(\bar{z}_{t-1}) - \mathcal{U}(z_{m,\tau_{m,t}}) - c_{m,t}(\theta), \tag{23}$$

  where $f$ is defined by

$$f(x) \stackrel{\Delta}{=} \mathcal{U}(-L(x, D_{\text{test}})). \tag{24}$$

  Return $\theta_t \leftarrow \theta_{t-1} + \eta \cdot \nabla_\theta O_t(\theta_{t-1})$

**ClientStrategy** $(\mathcal{P}_t, z_{m,\tau_{m,t}}, \bar{z}_{t-1})$**:**

  Return $b_{m,t} \stackrel{\Delta}{=} \mathbb{1}_{\delta_{m,t} > 0}$, where $\delta_{m,t}$ is the same as in (23) but with $\theta = \theta_t$

---

these unwanted clients. To demonstrate this intuition, we consider an extreme case where there exist Byzantine attacks carried out by malicious or faulty clients (Shi et al., 2022). We then demonstrate the robustness of the incentivized FL (labeled as "ICL" in the results) against two types of Byzantine attacks (Marano et al., 2008; Fang et al., 2020): random modification, which is a training data-based attack (Xia et al., 2019; Shi et al., 2022), and label flipping, which is a parameter-based attack (Jebreel et al., 2022; Gill et al., 2023). Specifically, the random modification is based on

generating local model parameters from a uniform distribution between $-0.25$ and $0.25$, and the label flipping uses cyclic alterations, such as changing 'dog' to 'cat', and vice versa. Byzantine client ratios are adjusted to two levels: $\{0.2, 0.3\}$. In the reported results below, *Byzantine-0.2* refers to a scenario where 20% of the total clients are adversarial. We generate 100 clients using IID partitioning of the training part of each dataset. Among participating clients, the server will select 10% of clients as active clients per round. We adopt the model architecture and hyperparameter settings similar to (Li et al., 2022), which are summarized in Tables 3 and 4.

Table 3: Network architecture

| Data | FashionMNIST | CIFAR10 | CINIC10 |
|---|---|---|---|
| CNN Hidden Size | | [120, 84] | |

Table 4: Hyperparameters

| Global | Communication round | 100 |
|---|---|---|
| Client | Batch size | 10 |
| | Local epoch | 5 |
| | Optimizer | SGD |
| | Learning rate | 3.00E-02 |
| | Weight decay | 5.00E-04 |
| | Momentum | 9.00E-01 |
| | Nesterov | ✓ |
| | Scheduler | Cosine Annealing |
| Objective function | Learning rate | 1.00E-04 |
| | Optimizer | SGD |
| | Learning rate | 1.00E-02 |
| | Weight decay | 5.00E-04 |
| | Momentum | 5.00E-01 |
| | Nesterov | ✓ |
| | Scheduler | Cosine Annealing |
| | Sigmoid $s$ | 0.005 |

**Pricing**  We consider the pricing plan in the form of (18) with

$$c_j = A_1(z; \theta_1) + \mathbb{1}_{j \in \mathbb{I}_A} A_2(z - z_m; \theta_2) \tag{25}$$

$$= \theta_1 \cdot z + \mathbb{1}_{j \in \mathbb{I}_A} \theta_1 \cdot z \cdot (-1 + \gamma \sigma_s(z - z_m - \theta_2)). \tag{26}$$

Accordingly, we replace the calculation of cost in (22) with

$$c_{m,t}(\theta) \triangleq \theta_1 \cdot \bar{z}_{\tau_{m,t}} \left( 1 + \rho \cdot (-1 + \gamma \sigma_s(\bar{z}_{\tau_{m,t}} - z_{m,\tau_{m,t}} - \theta_2)) \right) \tag{27}$$

In our study, we conduct an ablation test with three different values of the hyperparameter $\gamma$: 11, 101, and 2001. These are henceforth referred to as ICL plans 1, 2, and 3 in the following results. A larger value of $\gamma$ implies a more substantial penalty for the underperforming clients. The parameter $\theta_2$ is initialized in each global communication round using the Jenks natural breaks technique (Jenks & Caspall, 1971) and optimized based on the objective function.

**Profit**  Recall that the profit of a client or the server consists of monetary profit from participation fees and gain-converted profit from collaboration gains. In FL, we define the collaboration gain $Z_t$ as the negative test loss of the updated server model at round $t$, so the larger, the better. More specifically,

$$\bar{z}_t = \mathcal{G}(x_{m,t}, m \in \mathbb{I}_A) = -L(\bar{x}_t, D_{\text{test}})$$

$$\triangleq - \sum_{(\text{input},\text{output}) \in D_{\text{test}}} \ell(\text{input}, \text{output}; \bar{x}_t),$$

where $\ell$ is the cross-entropy loss under the model parameterized by $\bar{x}_t$. Likewise, the local gain of a client $m$ is defined by

$$z_{m,t} = \mathcal{G}(x_{m,t}) \stackrel{\Delta}{=} -L(x_{m,t}, D_{\text{test}}) \tag{28}$$

$$\stackrel{\Delta}{=} - \sum_{(\text{input},\text{output}) \in D_{\text{test}}} \ell(\text{input}, \text{output}; x_{m,t}). \tag{29}$$

Notably, the test set only needs to be stored and operated by the server, and the clients only need to access their own historical gains and the server's gains. We use $\lambda = 0.1$ and $\mathcal{U} : z \mapsto z$ (the identity map) in the experiment.

**Results** We visualize the learning curves on each dataset in Figures 4, 5, and 6. The model performance is assessed using the top-1 accuracy on the test dataset. We also summarize the best model performance in Tables 5, 6, and 7, respectively.

The following key points can be drawn from the results. Firstly, in comparison with non-incentivized Federated Learning (FL) based on FedAvg, our proposed incentivized FL (denoted as "ICL") algorithm can deliver a higher and more rapidly increasing model performance (representing Collaboration Gain in our context), as defined in (7). Across all settings–including two types of adversarial attacks, two ratios of adversarial clients, and three datasets–ICL with pricing plan 3 consistently outperforms FedAvg by a significant margin. On the other hand, ICL with pricing plan 1 underperforms, which is expected as it only imposes a mild penalty on laggard/adversarial active clients. The results suggest that it is possible to significantly mitigate the influence of malicious clients by precluding them from participating in an FL round, given that the pricing penalty is sufficiently large. Furthermore, the figures also indicate that the random modification attack poses a more significant threat compared to the label flipping attack, making it particularly difficult for non-incentivized FedAvg to converge.

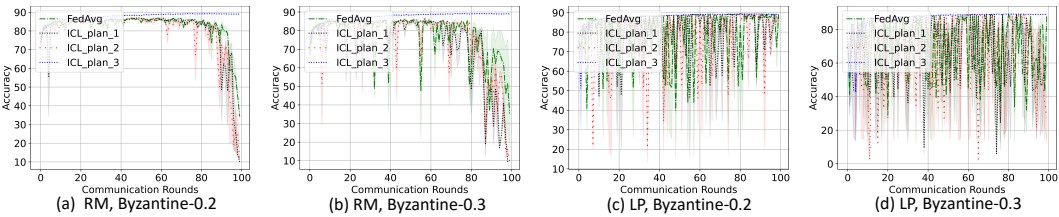

Figure 4: Learning curves of ICL (incorporating three pricing plans) and FedAvg measured by Accuracy, assessed for random modification (RM) label flipping (LP) attacks and two malicious client ratios (0.2 and 0.3), applied to the **FashionMNIST** dataset.

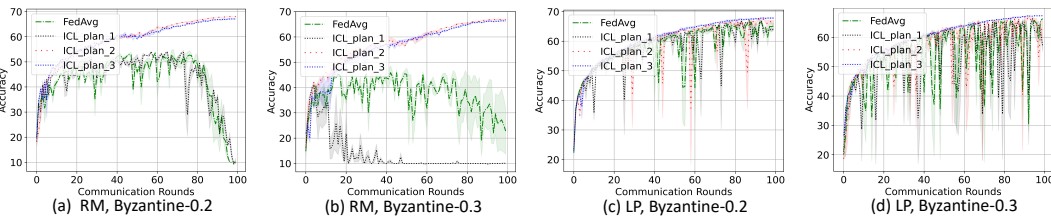

Figure 5: Learning curves of ICL (incorporating three pricing plans) and FedAvg measured by Accuracy, assessed for random modification (RM) label flipping (LP) attacks and two malicious client ratios (0.2 and 0.3), applied to the **CIFAR10** dataset.

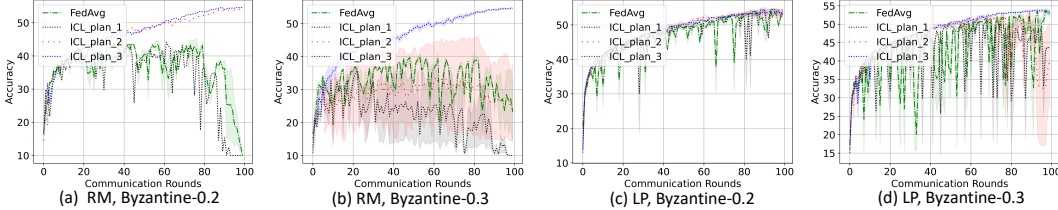

Figure 6: Learning curves of ICL (incorporating three pricing plans) and FedAvg measured by Accuracy, assessed for random modification (RM) label flipping (LP) attacks and two malicious client ratios (0.2 and 0.3), applied to the **CINIC10** dataset.

Table 5: Best model prediction accuracy of ICL (incorporating three pricing plans) and FedAvg measured by Accuracy, assessed for random modification (RM) label flipping (LP) attacks and two malicious client ratios (0.2 and 0.3), applied to the **FashionMNIST** dataset.

| Method | Random modification | | Label flipping | |
|---|---|---|---|---|
| | *Byzantine-0.2* | *Byzantine-0.3* | *Byzantine-0.2* | *Byzantine-0.3* |
| FedAvg | 87.1(0.1) | 86.5(0.1) | 89.6(0.0) | 89.3(0.0) |
| ICL pricing plan 1 | 87.2(0.0) | 86.4(0.0) | 89.5(0.1) | 89.0(0.1) |
| ICL pricing plan 2 | 87.0(0.0) | 86.4(0.3) | 89.3(0.1) | 89.4(0.2) |
| ICL pricing plan 3 | 89.4(0.0) | 89.2(0.1) | 89.4(0.1) | 89.2(0.1) |

Table 6: Best model prediction accuracy of ICL (incorporating three pricing plans) and FedAvg measured by Accuracy, assessed for random modification (RM) label flipping (LP) attacks and two malicious client ratios (0.2 and 0.3), applied to the **CIFAR10** dataset.

| Method | Random modification | | Label flipping | |
|---|---|---|---|---|
| | *Byzantine-0.2* | *Byzantine-0.3* | *Byzantine-0.2* | *Byzantine-0.3* |
| FedAvg | 55.7(0.4) | 50.1(0.3) | 67.3(0.2) | 66.9(0.2) |
| ICL pricing plan 1 | 54.8(0.2) | 42.0(0.9) | 67.0(0.4) | 66.9(0.3) |
| ICL pricing plan 2 | 68.1(0.0) | 67.0(0.2) | 67.6(0.1) | 67.1(0.1) |
| ICL pricing plan 3 | 67.1(0.2) | 66.4(0.1) | 67.9(0.2) | 67.4(0.1) |

Table 7: Best model prediction accuracy of ICL (incorporating three pricing plans) and FedAvg measured by Accuracy, assessed for random modification (RM) label flipping (LP) attacks and two malicious client ratios (0.2 and 0.3), applied to the **CINIC10** dataset.

| Method | Random modification | | Label flipping | |
|---|---|---|---|---|
| | *Byzantine-0.2* | *Byzantine-0.3* | *Byzantine-0.2* | *Byzantine-0.3* |
| FedAvg | 45.9(0.8) | 41.0(0.6) | 54.3(0.4) | 53.8(0.8) |
| ICL pricing plan 1 | 44.1(0.0) | 38.6(2.1) | 54.4(0.3) | 53.9(0.1) |
| ICL pricing plan 2 | 54.1(0.0) | 41.2(7.8) | 55.4(0.8) | 54.7(0.6) |
| ICL pricing plan 3 | 54.6(0.0) | 54.7(0.3) | 55.1(0.6) | 54.2(0.3) |

# E ICL FOR ASSISTED LEARNING

**Assisted learning (AL):** AL (Xian et al., 2020b) is a decentralized learning framework that enables autonomous organizations to significantly improve their performance by seeking feedback from peers with just a few assistance rounds, without sharing their private models, objectives, or data. Since then, AL has been expanded to encompass a broader range of learning tasks, including supervised learning (Diao et al., 2022b), recommendation systems (Diao et al., 2022d), and multi-task learning (Wang et al., 2022). AL has primarily focused on vertically partitioned data, where entities possess data with distinct feature variables collected from the same cohort of subjects. Recently, AL has been extended to support organizations with horizontally partitioned data based on the idea of transmitting model training trajectories, within applications to reinforcement learning (Chen et al., 2023b) (in which the assisting agent has diverse environments) and unsupervised domain adaptation (Chen et al., 2023a) (where the assisting agent possesses supplementary data domains).

In this section, we give a more specific incentivized AL setup and operational algorithm based on the development in Section 3.1 and the parallel assisted learning (PAL) (Wang et al., 2022) framework.

## E.1 OPERATIONAL ALGORITHM

**Review of non-incentivized PAL**. The main idea is to let each entity seek assistance from others by iteratively exchanging non-private statistics. Such statistics received by an entity will contain unknown task information deliberately injected by others so that each entity has an incentive to 'do well.' In the *training stage* (Stage I), at the first round of assistance, A sends its residual vector to B. Upon receipt of the query, B blends the received vector with some statistics $s_{b,a} \in \mathbb{R}^n$ calculated from its local task, treats it as learning labels, and fits a model. After that, B sends the residual back to A, who will initialize the next round. The above procedure continues until an appropriate stop criterion is triggered. The training stage initialized by A is then completed. After that, B will initialize another training stage by swapping the role with A. After the second training stage is completed, both entities can proceed to the prediction stage.

In the *prediction stage* (Stage II), upon arrival of a new feature vector, A queries the prediction results from B's local models and combines them with its own to form the final prediction. So does B. Each entity will need to tell how they blend the learned labels to 'decode' the faithful prediction values.

**Our incentivized PAL protocol**. Following Equation (14), each entity $i$ needs to evaluate $(u - c_i)\mu_{i,j} + c_j\mu_{j\leftarrow i}$ at a particular PAL round. We will estimate $\mu_{i,j}$ (respectively $\mu_{j\leftarrow i}$) with the collaboration gain from $i, j$ (respectively the additional gain brought by $i$ to $j$, at the last round they collaborated. More specifically, $\mu_{i,j}$ is estimated by the reduced prediction loss on a validation dataset for entity $i$'s task, comparing the collaborative PAL protocol trained from previous and current rounds. The $\mu_{j\leftarrow i}$ is estimated by the reduced prediction loss comparing the entity $i$'s locally trained model and the collaboratively trained PAL protocol at the current rounds.

The pseudocode is summarized in Algorithm 2. More specifically, the PAL training stage consists of multiple rounds. In each round, each of the three entities will decide who to collaborate based on the estimated expected gain. If a pair of entities favors each other, they will form PAL, and the remaining entity learns on its own; otherwise, every entity will learn on its own. The PAL training of any entity can be terminated at the discretion of that entity. This provides flexibility to ensure that any entity can stop appropriately if other participants' model/data are no longer helpful or its validation performance has reached a plateau. Despite when an entity exits the training stage, it will still be dedicated to working with others in the prediction stage for joint prediction, using its locally learned models.

## E.2 EXPERIMENTAL DETAILS FOR INCENTIVIZED AL

In the experimental studies, we considered the following setup.

**Data**  We consider a real-world clinical dataset MIMIC (Johnson et al., 2016) under PAL setting (Wang et al., 2022). We conduct experiments to show our pricing plan can promote mutually beneficial collaboration. We pre-processed MIMIC3 following (Purushotham et al., 2018; Harutyunyan et al., 2019) and obtained a dataset of 6916 patients and 18 variables. We used 35% training, 15% validation, and 50% testing. Suppose there are three divisions/entities that collected various

---

**Algorithm 2** Incentivized Parallel Assisted Learning

---

**Input:** Entity $i$ ($i \in [3]$) with local task label $y_i \in \mathbb{R}$, local feature variables $x_i \in \mathbb{R}^{d_i}$, concatenated predictor variables (in hindsight) $x \triangleq [x_1, x_2, x_3] \in \mathbb{R}^{d_1+d_2+d_3}$, pricing function $p_i : \Delta z \mapsto c_i \cdot \Delta z$, locally trained model $f_{i,0}$ (without collaboration), blending parameter $\tau_i$ for PAL, test loss function $L_i$, and Utility-Income function $\mathcal{U} : z \mapsto u \cdot z$

**Output:** Prediction protocol for each entity $i$ represented by $f_{i,t}(x_j, j \in \mathbb{C}_t)$, $t = 1, \ldots, T$, where $\mathbb{C}_t \subset [3]$ denotes the set of collaborating entities at round $t$.

**Initialization:** Entity $i$'s initial label $r_i \triangleq y_i - f_{i,0}(x_i)$, $i \in [3]$

**Training stage:**

    **for** each assistance round $t = 1, \ldots, T$ **do**

        **for** each entity $i \in [3]$ in parallel **do**

            For all $j \neq i$, let $\tau_{i,j,t}$ denotes the most recent round when $i, j$ collaborated by round $t$ (if any) and calculate:

$$q_{i,j,t} \triangleq \begin{cases} (u - c_i) \cdot \mu_{i,j,\tau_{i,j,t}} + c_j \cdot \mu_{j \leftarrow i, \tau_{i,j,t}} & \text{if } i, j \text{ have collaborated in at least one round} \\ \infty & \text{otherwise} \end{cases} \tag{30}$$

            Entity $i$ favors $i^* \triangleq \arg\max_{j \neq i} q_{i,j,t}$ – if there are more than one maximum, randomly choose one

            **if** *Entity $i$ has collaborated with entity $j$ and $\mu_{i,j,\tau_{i,j,t}} \leq 0$ for all $j \neq i$* **then**

                Entity $i$ will no longer favor anyone and stop collaborations

            **end**

        **end**

        **if** *there exists two entities, say A and B, that favor each other* **then**

            $f_{A,t}$, $\mu_{A,B}$, $\mu_{B \leftarrow A}$, $f_{B,t}$, $\mu_{B,A}$, $\mu_{A \leftarrow B}$ = **runPAL**(subject-aligned observations of $(x_A, r_A)$, $(x_B, r_B)$)

            $r_A \leftarrow y_A - f_{A,t}(x)$, $r_B \leftarrow y_B - f_{B,t}(x)$, $\mathbb{C}_t \leftarrow \{A, B\}$

        **end**

        For each remaining entity, say C, train a local model from $(x_C, r_C)$ to obtain a function $h_{C,t}(x_C)$, and construct a prediction-stage function $f_{C,t}(x) = f_{C,t-1}(x) + h_{C,t}(x_C)$.

    **end**

    For each entity, if its validation loss no longer decreases, it will quit the training stage and will be in the prediction stage

**runPAL** (Entity A's local sample of $(x_A, r_A)$, entity B's local sample of $(x_B, r_B)$)**:**

    Entities A and B run PAL for one round with $(x_a, r_a)$ and $(x_b, r_b)$, which result in two local models $h_{A,t}$ and $h_{B,t}$

    Entity A obtains a prediction-stage function $f_{A,t}(x) \triangleq f_{A,t-1}(x) + h_{A,t}(x_A) + h_{B,t}(x_B)$

    Entity A trains a local model from $(x_A, r_A)$ to obtain a function $\tilde{h}_{A,t}(x_A)$, and construct a a prediction-stage function without collaboration at $t$: $\tilde{f}_{A,t}(x) = f_{A,t-1}(x) + \tilde{h}(x_A)$.

    Entity A evaluates the gains using the empirical expectation over A's local test data (denoted by $\mathbb{E}_{\text{test}}$):

$$\mu_{A,B,t} \triangleq \mathbb{E}_{\text{test}}\{L_A(f_{A,t-1}(x), y_A) - L_A(f_{A,t}(x), y_A)\}, \tag{31}$$

$$\mu_{A \leftarrow B,t} \triangleq \mathbb{E}_{\text{test}}\{L_A(\tilde{f}_{A,t}(x), y_A) - L_A(f_{A,t}(x), y_A)\}.$$

    Repeat the above for entity B

    Return: A holds $f_{A,t}$, $\mu_{A,B,t}$, $\mu_{B \leftarrow A,t}$, B holds $f_{B,t}$, $\mu_{B,A,t}$, $\mu_{A \leftarrow B,t}$

**Prediction stage** (a future subject whose modality $x_i^{\text{f}}$ is observed by entity $i \subset [3]$)**:**

    Each entity $i$ will predict with $y_i^{\text{f}} = \sum_{t \in [T]: i \in \mathbb{C}_t} f_{i,t}(x_j, j \in \mathbb{C}_t)$

    **// Note**: each entity will only operate on its local data and receive assistance (namely $h_{j,t}(x_j)$) from other entities

---

features from the same cohort of patients and have different regression tasks. In particular, Entity 1, 2, and 3 aim to predict the heart rate, the systolic blood pressure, and the length of stay, respectively.

We supposed Entity 1 holds the variables of capillary refill rate, diastolic blood pressure, fraction inspired oxygen, heart rate, height, mean blood pressure, oxygen saturation, and respiratory rate; Entity 2 holds the variables of capillary refill rate, diastolic blood pressure, and fraction inspired oxygen. Entity 3 holds the variables of temperature, weight, pH, glucose, Glasgow coma scale total, Glasgow coma scale, eye Glasgow coma scale motor, and Glasgow coma scale verbal. Each entity will locally used a random forest model with 50 trees and max tree depth 5 in each round. We ran ten times of the experiments and recorded the standard errors, where the randomness comes from the local training and data splitting.

To demonstrate how incentives can help assisted learning, we studied the following three settings. In Setting $i$ ($i = 1, 2, 3$), entity $i$ uses $c_i = 0$, namely it does not pay, while other two entities use $c_j = u = 10$ (following the notation in Algorithm 2). It is expected that in Setting $i$, entity $i$ will not receive a good assistance from others as it does not incentivize peer entities.

**Results** The results are summarized in Fig. 7. The results show that for every entity $i$, its performance curves tends to decrease faster, or gain more, if it provide incentives to other peers. We also summarized the eventual PAL gain of each entity in Table 8. The results show that incentivized PAL can significantly improve upon an entity's local learning performance (without collaboration).

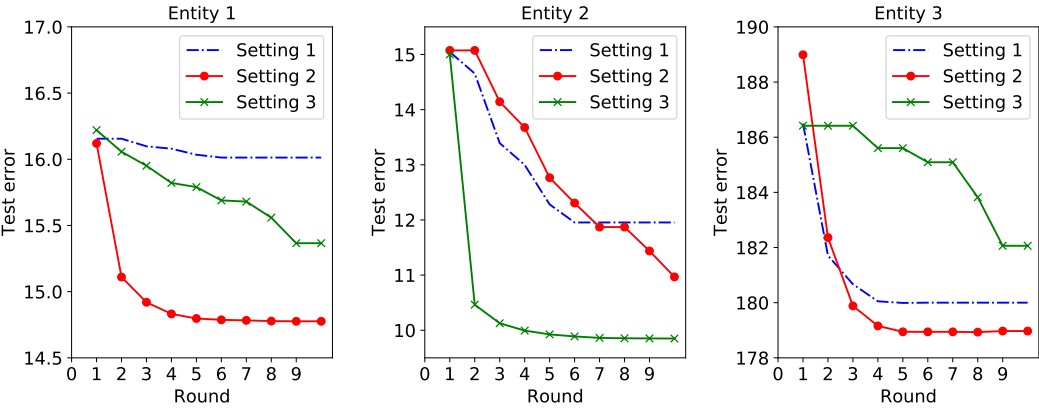

Figure 7: Test performance of each entity in incentivized PAL: test error (measured by quadratic loss) against assistance round in three settings. In Setting $i$, entity $i$ does not pay, namely $c_i = 0$ following the notation in Algorithm 2.

Table 8: Test performance of each entity after incentivized PAL in three settings.

| | Entity 1 | | | Entity 2 | | | Entity 3 | | |
|---|---|---|---|---|---|---|---|---|---|
| | Setting 1 | Setting 2 | Setting 3 | Setting 1 | Setting 2 | Setting 3 | Setting 1 | Setting 2 | Setting 3 |
| Incentivized PAL | 16.01 (0.05) | 14.78 (0.08) | 15.37 (0.09) | 11.95 (0.51) | 10.97 (0.46) | 9.85 (0.06) | 180.00 (1.16) | 178.97 (1.28) | 182.06 (2.14) |
| Local modeling | 16.22 (0.04) | | | 15.07 (0.05) | | | 188.99 (1.37) | | |

# F   ICL FOR MULTI-ARMED BANDIT

## F.1   COLLABORATIVE MULTI-ARMED BANDIT

As a follow-up to our Introduction section, we present another motivating example using MAB.

In the standard Multi-Armed Bandit (MAB) setting, suppose there are $M$ arms, each producing a gain $z_i$ with mean $\mu_i$. The player will select an arm, denoted by $i_t$, at each round and realize a gain, and its goal is to maximize the expected cumulative reward $\mathbb{E}\{\sum_{t=1}^{T} z_{i_t}\} = \mathbb{E}\{\sum_{t=1}^{T} \mu_{i_t}\}$ at any stopping time $T$. Note that the above $i_t$'s can be random due to the selection rule. Without loss of generality, we suppose the arms are ordered in such a way that $\mu_1 \geq \cdots \geq \mu_M$. We consider the following MAB-inspired collaborative setting, which we refer to as collaborative MAB.

*Collaborative MAB*: Each arm represents a candidate entity that knows its underlying $\mu_i$ but is unaware of others'. In each round, entities decide whether to participate and if so, they may be selected to realize the reward. All participants share the realized reward and pay accordingly. Meanwhile, the coordinator employs a selection scheme to maximize system-level profit.

We will demonstrate some key insights using the collaborative MAB problem. In particular, we will show that the collaboration requires screening at participation and selection stages, so that strong candidates will remain to maximize the collaboration gain, medium candidates will pay to enjoy the gain, and the laggard candidates (with low expected reward) will not participate in the game to avoid potential large costs.

## F.2   APPLICATION OF ICL TO THE MULTI-ARMED BANDIT

Specifically, we consider the selection rule that each time, the coordinator will select the entity with the largest average realized reward from history with probability $1 - \varepsilon$ and select among the participants uniformly at random with probability $\varepsilon$. We suppose $\mathcal{U}(z) = z$.

Then, the Equilibrium condition in Theorem 1 can be rewritten as

$$\text{Incent}_m : \ \mathbb{E}\{-c_m + \mu_A\} - \mu_m \geq 0, \quad \text{Incent}_{\text{sys}} : \ \mathbb{E}\{\lambda c_m + \mu_A - \mu_{A^{(-m)}}\} \geq 0, \tag{32}$$

where A and $A^{(-m)}$ denote respectively the selected active entity/arm from all the participants and that from all the participants excluding $m$.

Based on the above, we will then focus on the ideal equilibrium scenario where the best-performing arm, namely arm 1, consistently participates and also has the largest empirical reward in history, namely $\mathcal{A}$ equals 1 with probability $1 - \varepsilon$ and any remaining arm with probability $\varepsilon/(|\mathbb{I}_P| - 1)$. Then, we have

$$\mathbb{E}(\mu_A) = (1 - \varepsilon)\mu_1 + \varepsilon \frac{\sum_{i \in \mathbb{I}_P} \mu_i}{|\mathbb{I}_P|}, \tag{33}$$

$$\mathbb{E}(\mu_{A^{(-m)}}) = (1 - \varepsilon)\mu_1 + \varepsilon \frac{\sum_{i \in \mathbb{I}_P^{(-m)}} \mu_i}{|\mathbb{I}_P^{(-m)}|}, \ \forall m \neq 1, \tag{34}$$

$$\mathbb{E}(\mu_{A^{(-m)}}) - \mathbb{E}(\mu_A) \approx \varepsilon \frac{\bar{\mu}_{\mathbb{I}_P} - \mu_m}{|\mathbb{I}_P|}, \tag{35}$$

where the last equation is an asymptotic approximation for large $|\mathbb{I}_P|$, and $\bar{\mu}_{\mathbb{I}_P} = \frac{\sum_{i \in \mathbb{I}_P} \mu_i}{|\mathbb{I}_P|}$ is the average of $\mu$'s within $\mathbb{I}_P$. Assume $\lambda > 0$ for now. Taking the above to (32), we obtain the conditions for the pricing plan:

$$\mathbb{E}\{c_m\} \leq (1 - \varepsilon)\mu_1 + \varepsilon \mathbb{E}\{\bar{\mu}_{\mathbb{I}_P}\} - \mu_m, \tag{36}$$

$$\mathbb{E}\{c_m\} \geq \frac{\varepsilon}{\lambda |\mathbb{I}_P|}(\bar{\mu}_{\mathbb{I}_P} - \mu_m). \tag{37}$$

● Interpretation of (36): Arm $m$ will participate if its expected cost is no larger than the expected additional reward. For large $T$, we often let $\varepsilon$ vanish. In this case, we can further approximate this condition as

$$\mathbb{E}\{c_m\} \leq \mu_1 - \mu_m. \tag{38}$$

• Interpretation of (37): The system will welcome Arm $m$ if its expected payment can overcome its harm in degrading the collaboration reward (as reflected in the term $\bar{\mu}_{\mathbb{I}_p} - \mu_m$). Also, the large $\lambda$, the more tolerance of the system to the "laggard" arms. This is intuitive as a large $\lambda$ means the system's profit from participation costs outweighs that from the collaboration gain.

**Remark 7** (Incentive eliminates the "laggard" candidates)**.** In practice, there may be many laggard or even adversarial arms, meaning that their $\mu_m$'s are much smaller than $\bar{\mu}_{\mathbb{I}_p}$, that aim to participate to enjoy the gain or harm the system with only a small participation cost. To alleviate this concern, we should design the pricing plan in a way that significantly penalize the laggard arms, so that when they are selected to be active, albeit with a small probability, the overall expected participation cost is very high. With such an incentive design, we aim to achieve the following goals:

1) prevent the laggard arms from participating by leveraging their incentives, which can further improve the collaboration gain,

2) encourage the medium-performing arms to participate to enjoy the collaboration gain, which can boost the overall profit, and

3) reward the top-performing arms to participate to make sure that they can be consistently selected to be active.

Next, we will exemplify the above insights by studying a specific pricing plan.

**Example**: Consider the pricing plan

$$c_j = b_0 + b_1 \mathbb{1}_{z_j < \kappa_1} - b_2 \mathbb{1}_{z_j > \kappa_2} \tag{39}$$

where $b_0 \geq 0$ is the baseline price, $b_1 \geq 0$ is the extra price for realizing a reward (if selected) that is less than a threshold $\kappa_1$, and $b_2 \geq 0$ is the reward for realizing a reward that is greater than a threshold $\kappa_2$.

We suppose each arm $m$ knows its distribution of reward $z_m$ and the mean $\mu_m$. Assume $Z_m \mid \mu_m \sim \mathcal{N}(\mu_m, s^2)$, and let $F$ denote the cumulative distribution function (CDF) of $\mathcal{N}(0, s^2)$. According to (36), arm $m$'s condition to participate is

$$\mathbb{E}\{c_m\} \leq (1 - \varepsilon)\mu_1 + \varepsilon \mathbb{E}\{\bar{\mu}_{\mathbb{I}_p}\} - \mu_m,$$

which we approximate with

$$
\begin{aligned}
& b_0 + b_1 \mathbb{P}(z_m < \kappa_1) - b_2 \mathbb{P}(z_m > \kappa_2) \\
&= b_0 + b_1 F_s(\kappa_1 - \mu_m) - b_2 \mathcal{F}_s(\mu_m - \kappa_2) \\
&\leq (1 - \varepsilon) \max_{j \in [M]} \hat{\mu}_{j,t} + \varepsilon \cdot M^{-1} \sum_{j \in [M]} \hat{\mu}_{j,t} - \mu_m,
\end{aligned}
\tag{40}
$$

where $\hat{\mu}_{j,t}$ is the empirical average reward realized by arm $j$ up to time $t$.

To compare with the standard non-incentivized MAB, we will consider $\lambda = 0$, namely the system-level profit is simply the cumulative realized rewards. We will show that **with incentive design**, the MAB can attain a cumulative consistently larger that that of a non-incentivized MAB. To that end, we discuss how to choose the pricing plan as parameterized by $\boldsymbol{b} = [b_0, b_1, b_2]$ and $\boldsymbol{\kappa} = [\kappa_1, \kappa_2]$. They need to satisfy two conditions:

1) **Effective incentive**: The arms incentivized to participate must be of high quality (with large means) so that the exploration cost of MAB can be significantly reduced to benefit the cumulative rewards.

In the asymptotic regime where the number of rounds is large and $\varepsilon$ tends to zero, condition (40) becomes

$$b_0 + b_1 F_s(\kappa_1 - \mu_m) - b_2 \mathcal{F}_s(\mu_m - \kappa_2) \leq \mu_1 - \mu_m. \tag{41}$$

It can be seen that both the left and right-hand sides are decreasing in $\mu_m$. We will design $\boldsymbol{b}, \boldsymbol{\kappa}$ such that

$$\mu_m \mapsto \mu_1 - \mu_m - \left\{ b_0 + b_1 F_s(\kappa_1 - \mu_m) - b_2 \mathcal{F}_s(\mu_m - \kappa_2) \right\}, \tag{42}$$

referred to as the *profit-performance function*, is strictly increasing in $\mu_m$. Recall that $\mathbb{I}_P$ is the set of $m$ that meets the condition (41). Therefore, $\mathbb{I}_P$ can be written in the form of

$$\mathbb{I}_P = \{m : \mu_m \geq \mu_{m^*}\} \tag{43}$$

where $m^*$ is such that $\mu_{m^*}$ is the smallest among all $\mu$'s that guarantees the condition (41), namely the arms whose underlying $\mu$ is above a threshold.

2) **Non-negative balance**: The overall participation cost should be non-negative so that the system does not need to spend extra cost on incentivizing candidate arms.

For this condition to hold, we should have

$$\sum_{m \in \mathbb{I}_P} \{b_0 + b_1 F_s(\kappa_1 - \mu_m) - b_2 \mathcal{F}_s(\mu_m - \kappa_2)\} \geq 0$$

where $\mathbb{I}_P$ is determined by the condition in (43).

We conduct a simulated experiment to demonstrate the insights. We generate $M$ $\mu$'s from a Gaussian distribution, generate $z$'s from Gaussian distributions centered at the corresponding $\mu$, run $T$ time steps. We choose $M = 50, T = 150$, and hyperparameters $\boldsymbol{b} = [1, 5, 10], \boldsymbol{\kappa} = [2, 4]$. We record the system's cumulative profit and the participation activities of each arm over 20 random experiments and plot their average in Fig. 8. To show the pricing plan is reasonable, we also visualized the underlying $\mu$'s from one random experiment and its corresponding profit-performance function as defined in (42).

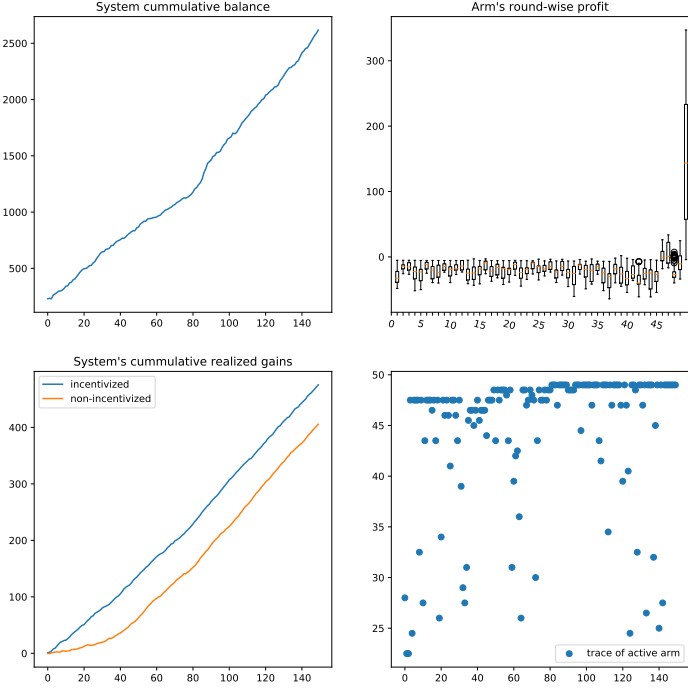

Figure 8: Performance of the incentivized MAB: system's cumulative profit (from participation costs), 50 arms' profit in boxplot, cumulative realized rewards and comparison with nonincentivized game, and trace of selected active arms (larger index, larger performance), averaged over 20 random experiments.

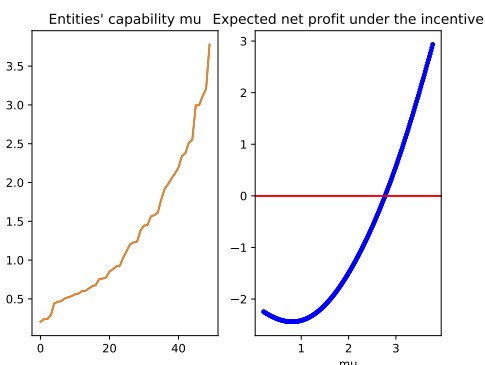

Figure 9: Illustration of a particular incentive ($b = [1, 5, 10], \kappa = [2, 4]$): an arm's expected net profit by participating the game against its underlying capability ($\mu$). Arms with better performance expect to have positive profits and are therefore incentivized to participate.

# G    PROOFS OF THEORETICAL RESULTS

## G.1    PROOF OF PROPOSITION 1

**Proposition 1 (restatement)** Let $\lambda' \triangleq \frac{\lambda-1}{|\mathbb{I}_P|+1}$. The Objective (5) where $\lambda'$ is replaced with $\lambda$ is equivalent to maximizing the average social welfare defined by $(\text{PROFIT}_{\text{sys}} + \sum_{m \in \mathbb{I}_P} \text{PROFIT}_m)/(|\mathbb{I}_P|+1)$.

*Proof.* According to (2) and (3), and the fact that non-participating candidates have a zero profit from collaboration, we have

$$
\text{PROFIT}_{\text{sys}} + \sum_{m \in \mathbb{I}_P} \text{PROFIT}_m
$$

$$
= \lambda \sum_{m \in \mathbb{I}_P} c_m + \mathcal{U}(z_{\mathbb{I}_A}) + \sum_{m \in \mathbb{I}_P} \mathbb{1}_{m \in \mathbb{I}_P} \cdot (-c_m + \mathcal{U}(z_{\mathbb{I}_A}) - \mathcal{U}(z_m))
$$

$$
= \lambda \sum_{m \in \mathbb{I}_P} c_m + \mathcal{U}(z_{\mathbb{I}_A}) + \sum_{m \in \mathbb{I}_P} \cdot (-c_m + \mathcal{U}(z_{\mathbb{I}_A}) - \mathcal{U}(z_m))
$$

$$
= (\lambda - 1) \sum_{m \in \mathbb{I}_P} c_m + (|\mathbb{I}_P| + 1)\mathcal{U}(z_{\mathbb{I}_A}) - \sum_{m \in \mathbb{I}_P} \mathcal{U}(z_m)
$$

$$
= (|\mathbb{I}_P| + 1) \times \left\{ \frac{\lambda - 1}{|\mathbb{I}_P| + 1} \sum_{m \in \mathbb{I}_P} c_m + \mathcal{U}(z_{\mathbb{I}_A}) - \text{const} \right\},
$$

where $\text{const} = (|\mathbb{I}_P| + 1)^{-1} \sum_{m \in \mathbb{I}_P} \mathcal{U}(z_m)$ is a term that does not depend on the incentive mechanism. Therefore, maximizing the average profit is equivalent to maximizing

$$
\frac{\lambda - 1}{|\mathbb{I}_P| + 1} \sum_{m \in \mathbb{I}_P} c_m + \mathcal{U}(z_{\mathbb{I}_A}), \tag{44}
$$

which concludes the proof.  $\square$

## G.2    PROOF OF THEOREM 1

**Theorem 1 (restatement)** The condition for a strategy profile $(b_m^*, m \in [M], \mathcal{P}^*, \mathcal{S}^*)$ to attain a Nash equilibrium is

$$
\text{Incent}_m : \mathbb{E}\{-c_m + \mathcal{U}(z_{\mathbb{I}_A}) - \mathcal{U}(z_m)\} \geq 0, \tag{45}
$$

$$
\text{Incent}_{\text{sys}} : \mathbb{E}\{\lambda c_m + \mathcal{U}(z_{\mathbb{I}_A}) - \mathcal{U}(z_{\mathbb{I}_A^{(-m)}})\} \geq 0, \tag{46}
$$

if and only if $m \in \mathbb{I}_P$.

*Proof.* From the perspective each entity $m$, its expected profit if participating the game is $\mathbb{E}\{-c_m + \mathcal{U}(z_{\mathbb{I}_A}) - \mathcal{U}(z_m)\}$. Thus, conditional on other parties' decisions, entity $m$ will participate if and only if Inequality 45 holds. From the perspective each entity $m$, from the definition in (3), the additional expected profit of a candidate $m$ participating the game is

$$
\mathbb{E}\left\{\lambda \sum_{i \in \mathbb{I}_P} c_i + \mathcal{U}(z_{\mathbb{I}_A})\right\} - \mathbb{E}\left\{\lambda \sum_{i \in \mathbb{I}_P - \{m\}} c_i + \mathcal{U}(z_{\mathbb{I}_A^{(-m)}})\right\} = \mathbb{E}\left\{\lambda c_m + \mathcal{U}(z_{\mathbb{I}_A}) - \mathcal{U}(z_{\mathbb{I}_A^{(-m)}})\right\}.
\tag{47}
$$

Then, the system's strategy is to welcome a participant $m$ if and only if its expected marginal gain brought by this participant is nonnegative, which leads to Condition 46.

$\square$

## G.3    PROOF OF THEOREM 2

**Theorem 2 (restatement)** Let $\bar{\zeta}_{\mathbb{I}_P}$ and $\bar{x}_{\mathbb{I}_P}$ denote all the participants' average of $\zeta$ and weighted average of $X$, namely

$$
\bar{\zeta}_{\mathbb{I}_P} \triangleq \frac{\sum_{i \in \mathbb{I}_P} \zeta_i}{|\mathbb{I}_P|}, \quad \bar{x}_{\mathbb{I}_P} \triangleq \frac{\sum_{i \in \mathbb{I}_P} \zeta_i x_i}{\sum_{i \in \mathbb{I}_P} \zeta_i}. \tag{48}
$$

Assume that $\max\{|\zeta_m|, \|\zeta_m x_m\|_\infty\} \le \tau$ for all $m \in \mathbb{I}_P$ and $\log d/|\mathbb{I}_P| \to 0$ as $|\mathbb{I}_P| \to \infty$. Then, the equilibrium Conditions (9) and (10) can be written as

$$\mathbb{E}\{c_m\} \le \mathbb{E}\left\{(\mathcal{U} \circ \mathcal{G})(\bar{x}_{\mathbb{I}_P} + o_p(1)) - \mathcal{U} \circ \mathcal{G}(x_m)\right\}, \tag{49}$$

$$\lambda \mathbb{E}\{c_m\} \ge \mathbb{E}\left\{(\mathcal{U} \circ \mathcal{G})'(\bar{x}_{\mathbb{I}_P} + o_p(1))\frac{b_m}{(1 + o_p(1))\rho}\frac{\zeta_m}{K\bar{\zeta}_{\mathbb{I}_P}}(\bar{x}_{\mathbb{I}_P} - x_m)\right\}, \tag{50}$$

where $o_p(1)$ denotes a term whose $\ell_1$-norm converges in probability to zero as $|\mathbb{I}_P|$ goes to infinity.

*Proof.* By the definition of $z_{\mathbb{I}_A}$ and Taylor expansion, we have

$$\mathcal{U}(z_{\mathbb{I}_A}) = \mathcal{U} \circ \mathcal{G}\left(\frac{\sum_{i \in \mathbb{I}_A} \zeta_i x_i}{\sum_{i \in \mathbb{I}_A} \zeta_i}\right) = \mathcal{U} \circ \mathcal{G}\left(\sum_{i \in \mathbb{I}_P} \frac{B_i \zeta_i x_i}{\sum_{j \in \mathbb{I}_P} B_j \zeta_j}\right),$$

$$\mathcal{U}(z_{\mathbb{I}_A^{(-m)}}) = \mathcal{U} \circ \mathcal{G}\left(\sum_{i \in \mathbb{I}_P - \{m\}} \frac{B_i \zeta_i x_i}{\sum_{j \in \mathbb{I}_P - \{m\}} B_j \zeta_j}\right)$$

$$= \mathcal{U}(z_{\mathbb{I}_A}) + (\mathcal{U} \circ \mathcal{G})'(\omega_{K,m}) \times \left(\sum_{i \in \mathbb{I}_P - \{m\}} B_i \zeta_i x_i\left(\frac{1}{\sum_{j \in \mathbb{I}_P - \{m\}} B_j \zeta_j} - \frac{1}{\sum_{j \in \mathbb{I}_P} B_j \zeta_j}\right) - \frac{b_m \zeta_m x_m}{\sum_{j \in \mathbb{I}_P} B_j \zeta_j}\right)$$

$$= \mathcal{U}(z_{\mathbb{I}_A}) + (\mathcal{U} \circ \mathcal{G})'(\omega_{K,m}) \times \frac{b_m \zeta_m}{\sum_{j \in \mathbb{I}_P} B_j \zeta_j}\left(\frac{\sum_{i \in \mathbb{I}_P - \{m\}} B_i \zeta_i x_i}{\sum_{j \in \mathbb{I}_P - \{m\}} B_j \zeta_j} - x_m\right), \tag{51}$$

where $\omega_{K,m}$ is on the line connecting

$$\frac{\sum_{i \in \mathbb{I}_P} B_i \zeta_i x_i}{\sum_{j \in \mathbb{I}_P} B_j \zeta_j} \quad \text{and} \quad \frac{\sum_{i \in \mathbb{I}_P - \{m\}} B_i \zeta_i x_i}{\sum_{j \in \mathbb{I}_P - \{m\}} B_j \zeta_j}. \tag{52}$$

When $K \triangleq |\mathbb{I}_P|$ is large, we will show that both the above terms (and thus $\omega_{K,m}$) will concentrate at $\bar{x}_{\mathbb{I}_P}$ with high probability. Let $x_\ell$ denote the $\ell$-th element of a vector $X \in \mathbb{R}^d$. Using the bound of $\|\zeta_i x_i\|_\infty$, the fact that $b_m$'s follow independent $B(\rho)$, and Hoeffding's inequality, we have for any small constant $v > 0$,

$$\mathbb{P}\left(\left|\sum_{i \in \mathbb{I}_P} B_i \zeta_i x_{i,\ell} - \rho \sum_{i \in \mathbb{I}_P} \zeta_i x_{i,\ell}\right| \ge vK\tau\bar{\zeta}_{\mathbb{I}_P}\right) \le 2\exp(-v^2\bar{\zeta}_{\mathbb{I}_P}^2 K) \tag{53}$$

for every $\ell = 1, \ldots, d$. It follows from the union bound that

$$\mathbb{P}\left(\left\|\sum_{i \in \mathbb{I}_P} B_i \zeta_i x_i - \rho \sum_{i \in \mathbb{I}_P} \zeta_i x_i\right\|_\infty \le vK\tau\bar{\zeta}_{\mathbb{I}_P}\right) \ge 1 - 2d\exp(-v^2\bar{\zeta}_{\mathbb{I}_P}^2 K). \tag{54}$$

Likewise, we have

$$\mathbb{P}\left(\left|\sum_{i \in \mathbb{I}_P} B_i \zeta_i - \rho \sum_{i \in \mathbb{I}_P} \zeta_i\right| = \left|\sum_{i \in \mathbb{I}_P} B_i \zeta_i - \rho K\bar{\zeta}_{\mathbb{I}_P}\right| \le vK\tau\bar{\zeta}_{\mathbb{I}_P}\right) \ge 1 - 2\exp(-v^2\bar{\zeta}_{\mathbb{I}_P}^2 K). \tag{55}$$

Taking Inequalities (54) and (55) into (52), we have

$$\left\|\frac{\sum_{i \in \mathbb{I}_P} B_i \zeta_i x_i}{\sum_{j \in \mathbb{I}_P} B_j \zeta_j} - \frac{\rho \sum_{i \in \mathbb{I}_P} \zeta_i x_i}{\rho \sum_{i \in \mathbb{I}_P} \zeta_i}\right\|_\infty$$

$$= \left\|\frac{\sum_{i \in \mathbb{I}_P} B_i \zeta_i x_i - \rho \sum_{i \in \mathbb{I}_P} \zeta_i x_i}{\rho K\bar{\zeta}_{\mathbb{I}_P}} + \left(\frac{1}{\sum_{j \in \mathbb{I}_P} B_j \zeta_j} - \frac{1}{\rho K\bar{\zeta}_{\mathbb{I}_P}}\right)\sum_{i \in \mathbb{I}_P} B_i \zeta_i x_i\right\|_\infty$$

$$\le \frac{vK\tau\bar{\zeta}_{\mathbb{I}_P}}{\rho K\bar{\zeta}_{\mathbb{I}_P}} + \frac{vK\tau\bar{\zeta}_{\mathbb{I}_P}}{(\rho K\bar{\zeta}_{\mathbb{I}_P})(\rho K\bar{\zeta}_{\mathbb{I}_P} - vK\tau\bar{\zeta}_{\mathbb{I}_P})}\left(\left\|\rho \sum_{i \in \mathbb{I}_P} \zeta_i x_i\right\|_\infty + vK\tau\bar{\zeta}_{\mathbb{I}_P}\right)$$

$$\le \frac{v\tau}{\rho} + \frac{v\tau}{\rho(\rho - v\tau)K\bar{\zeta}_{\mathbb{I}_P}}(\rho K\tau\bar{\zeta}_{\mathbb{I}_P} + vK\tau\bar{\zeta}_{\mathbb{I}_P}) = \frac{v\tau}{\rho} + \frac{v(\rho + v)\tau^2}{\rho(\rho - v\tau)} \tag{56}$$

where the last two lines are due to the triangle inequality. From (54), (55), and (56), and the assumption that $\log d/K \to 0$ as $K \to \infty$, we can choose $v = (K^{-1} \log d)^{2/3}$ so that $v \to 0$ and $v^2 \cdot K/\log d \to \infty$ as $K \to \infty$, which leads to

$$\frac{\sum_{i\in\mathbb{I}_P} B_i \zeta_i x_i}{\sum_{j\in\mathbb{I}_P} B_j \zeta_j} = \frac{\rho \sum_{i\in\mathbb{I}_P} \zeta_i x_i}{\rho \sum_{i\in\mathbb{I}_P} \zeta_i} + o_p(1) = \bar{x}_{\mathbb{I}_P} + o_p(1), \tag{57}$$

where $o_p(1)$ denotes a term whose $\ell_1$-norm converges in probability to zero. The same result applies to $\frac{\sum_{i\in\mathbb{I}_P - \{m\}} B_i \zeta_i x_i}{\sum_{j\in\mathbb{I}_P - \{m\}} B_j \zeta_j}$.

By a similar argument as above, we have

$$\frac{1}{\sum_{j\in\mathbb{I}_P} B_j \zeta_j} = \frac{1}{\rho K (\bar{\zeta}_{\mathbb{I}_P} + o_p(1))}. \tag{58}$$

Taking (57) and (58) into (51), we conclude the proof.

$\square$

### G.4 PROOF OF PROPOSITION 2

**Proposition 2 (restatement)** Suppose the information transmitted from participant $m$ is the mean and variance of $x_m \in \mathbb{R}$, denoted by $\mu_m, \sigma^2$ for all $m \in \mathbb{I}_P$. Assume the gain is defined by the quadratic loss $\mathcal{G}(x) \triangleq -\mathbb{E}(X - \mu)^2$, where $\mu$ represents the underlying parameter of interest, and the participants' weights $\zeta_m$'s are the same. Assume that $\sigma^2/(|\mathbb{I}_P| \cdot \max_{m\in\mathbb{I}_P}(\mu_m - \mu)^2) \to \infty$ as $|\mathbb{I}_P| \to \infty$. Then, we have $U(q)/U(q^*) \to_p 1$ as $|\mathbb{I}_P| \to \infty$.

*Proof.* By the definition of $\mathcal{G}$, we have

$$\begin{aligned}
\mathcal{G}(x_m, m \in \mathbb{I}_A) &= \mathcal{G}\left(\frac{\sum_{m\in\mathbb{I}_P} b_m x_m}{\sum_{m\in\mathbb{I}_P} b_m}\right) \\
&= -\mathbb{E}\left(\frac{\sum_{m\in\mathbb{I}_P} b_m x_m}{\sum_{m\in\mathbb{I}_P} b_m} - \mu\right)^2 \\
&= -(T_1 + T_2)
\end{aligned} \tag{59}$$

where

$$\begin{aligned}
T_1 &\triangleq \mathbb{E}\left(\frac{\sum_{m\in\mathbb{I}_P} b_m x_m}{\sum_{m\in\mathbb{I}_P} b_m} - \frac{\sum_{m\in\mathbb{I}_P} b_m \mu_m}{\sum_{m\in\mathbb{I}_P} b_m}\right)^2 \\
&= \mathbb{E}\left(\mathrm{Var}\left(\frac{\sum_{m\in\mathbb{I}_P} b_m x_m}{\sum_{m\in\mathbb{I}_P} b_m} \mid b_m, m \in \mathbb{I}_P\right)\right) \\
&= \mathbb{E}\left(\frac{\sum_{m\in\mathbb{I}_P} b_m^2 \mathrm{Var}(x_m)}{\left(\sum_{m\in\mathbb{I}_P} b_m\right)^2} \mid b_m, m \in \mathbb{I}_P\right) \\
&= \mathbb{E}\left(\frac{\sum_{m\in\mathbb{I}_P} b_m \sigma^2}{\left(\sum_{m\in\mathbb{I}_P} b_m\right)^2}\right) \\
&= \mathbb{E}\left(\frac{\sigma^2}{\sum_{m\in\mathbb{I}_P} b_m}\right) = \frac{|\mathbb{I}_P|^{-1} \sigma^2}{|\mathbb{I}_P|^{-1} \sum_{m\in\mathbb{I}_P} b_m}
\end{aligned} \tag{60}$$

$$T_2 \triangleq \mathbb{E}\left(\frac{\sum_{m\in\mathbb{I}_P} b_m \mu_m}{\sum_{m\in\mathbb{I}_P} b_m} - \mu\right)^2 \leq \mathbb{E}(\Delta\mu^2). \tag{61}$$

By the Markov inequality, we have

$$
\begin{aligned}
\mathbb{P}\left(\left|\frac{\sum_{m\in\mathbb{I}_{\mathrm{P}}} b_m}{|\mathbb{I}_{\mathrm{P}}|} - \rho\right| \geq \varepsilon\right) &= \mathbb{P}\left(\left|\frac{\sum_{m\in\mathbb{I}_{\mathrm{P}}}(b_m - q_m)}{|\mathbb{I}_{\mathrm{P}}|}\right| \geq \varepsilon\right) \\
&\leq \varepsilon^{-2}\mathbb{E}\left|\frac{\sum_{m\in\mathbb{I}_{\mathrm{P}}}(b_m - q_m)}{|\mathbb{I}_{\mathrm{P}}|}\right|^2 \\
&= \varepsilon^{-2}\frac{\sum_{m\in\mathbb{I}_{\mathrm{P}}} q_m(1 - q_m)}{|\mathbb{I}_{\mathrm{P}}|^2} \\
&\leq \varepsilon^{-2}\frac{1}{4|\mathbb{I}_{\mathrm{P}}|} \to 0
\end{aligned}
$$

as $|\mathbb{I}_{\mathrm{P}}| \to \infty$. Therefore, based on (60), (61), and the assumption that $|\mathbb{I}_{\mathrm{P}}|^{-1}\sigma^2 \gg \Delta\mu^2$, $T_1$ will asymptotically dominate $T_2$ and

$$
\frac{\mathcal{G}(x_m, m \in \mathbb{I}_{\mathrm{A}})}{(\rho|\mathbb{I}_{\mathrm{P}}|)^{-1}\sigma^2} \to_p 1 \tag{62}
$$

as $|\mathbb{I}_{\mathrm{P}}| \to \infty$. Since the denominator in (62) does not depend on the particular $q$ introduced in (11), we conclude the proof. $\qquad\square$

### G.5   Proof of Theorem 3

We consider a more general setting. Let $p_i(\Delta z)$ denote the price $i$ is willing to pay for any additional gain of $\Delta z$. The actual gain entity $i$ will pay in a collaboration, say with $j$, is denoted by $C_{i\leftarrow j} \overset{\Delta}{=} p_i(z_{i,j} - z_i) - p_j(z_{j,i} - z_j)$. Let $\mu_i = \mathbb{E}\{z_i\}$ and $\mu_{i,j} = \mathbb{E}\{Z\}_{i,j}$ for all $i, j \in [M]$. For all $i, j$, let $\mu_{i\leftarrow j} \overset{\Delta}{=} \mu_{i,j} - \mu_i$ denote the expected additional gain bought by entity $i$ to $j$.

**Theorem 3 (restatement)** Suppose $p_i(\Delta z) = c_i \cdot \Delta z$ for all $i \in [M]$ and $\mathcal{U}(z) = u \cdot z$. Then, there exists a consensus on collaboration if and only if there exist two entities, say $i, j \in [M]$, such that

$$
\mathbb{E}\{\mathcal{U}(z_{i,j}) - C_{i\leftarrow j}\} \geq 0. \tag{63}
$$

In the linear case, the above conditions became

$$
\begin{aligned}
(u - c_i)\mu_{i,j} + c_j\mu_{j\leftarrow i} &\geq (u - c_i)\mu_{i,k} + c_k\mu_{k\leftarrow i}, \\
(u - c_j)\mu_{j,i} + c_i\mu_{i\leftarrow j} &\geq (u - c_j)\mu_{j,k} + c_k\mu_{k\leftarrow j},
\end{aligned}
$$

for all $k \neq i, j$.

*Proof.* Based on the setup, the expected profit of an entity $i$ will be the sum of the profit converted from PAL gain minus the paid cost. We let $\mathrm{PROFIT}_{i\leftarrow j}$ and $z_{i,j}$ respectively denote the $\mathrm{PROFIT}_i$ and the collaboration gain $Z$ generated from the PAL formed by entities $i$ and $j$. Then, we have

$$
\mathrm{PROFIT}_{i\leftarrow j} \overset{\Delta}{=} \mathbb{E}\{\mathcal{U}(z_{i,j}) - C_{i\leftarrow j}\}. \tag{64}
$$

For entity $i$ to favor $j$ over others, we have

$$
\mathrm{PROFIT}_{i\leftarrow j} \geq \mathrm{PROFIT}_{i\leftarrow k}, \quad \forall k \neq i, j, \tag{65}
$$

which, according to (64), implies that

$$
\mathbb{E}\{\mathcal{U}(z_{i,j})\} - \mathbb{E}\{C_{i\leftarrow j}\} \geq \mathbb{E}\{\mathcal{U}(z_{i,k})\} - \mathbb{E}\{C_{i\leftarrow k}\}, \quad \forall k \neq i, j. \tag{66}
$$

Likewise, for entity $j$ to favor $i$ over others, we have

$$
\mathbb{E}\{\mathcal{U}(z_{j,i})\} - \mathbb{E}\{C_{j\leftarrow i}\} \geq \mathbb{E}\{\mathcal{U}(z_{j,k})\} - \mathbb{E}\{C_{j\leftarrow k}\}, \quad \forall k \neq i, j. \tag{67}
$$

Finally, for entity $i$ to have a non-negative profit so that it will participate in the collaboration, we have

$$
\mathbb{E}\{\mathcal{U}(z_{i,j})\} - \mathbb{E}\{C_{i\leftarrow j}\} \geq 0. \tag{68}
$$

These proves the first part of the theorem.

With the linear assumption of $p_i$'s, we have

$$\mathbb{E}\{\mathcal{U}(z_{i,j})\} - \mathbb{E}\{C_{i \leftarrow j}\} = u\mu_{i,j} - \{c_i(\mu_{i,j} - \mu_i) - c_j(\mu_{i,j} - \mu_j)\} \tag{69}$$

$$= (u - c_i + c_j)\mu_{i,j} + c_i\mu_i - c_j\mu_j, \tag{70}$$

and similarly for other subscripts. Thus, it can be calculated that Inequalities (66) and (67) can reduce to

$$(u - c_i)(\mu_{i,j} - \mu_{i,k}) + c_j\mu_{j \leftarrow i} - c_k\mu_{k \leftarrow i} \geq 0,$$
$$(u - c_j)(\mu_{j,i} - \mu_{j,k}) + c_i\mu_{i \leftarrow j} - c_k\mu_{k \leftarrow j} \geq 0, \quad \forall k \neq i, j.$$

Also, Inequality (68) reduces to

$$u\mu_{i,j} - c_i\mu_{i \leftarrow j} + c_j\mu_{j \leftarrow i} \geq 0. \tag{71}$$

These conclude the proof. □

### G.6   PROOF OF THEOREM 4

**Theorem 4 (restatement)** Assume $\mathcal{U}$ is a pre-specified non-decreasing function. Consider the pricing plan in (15) where $C$ can be any non-negative and non-decreasing function. Let $\mu_m \triangleq \mathbb{E}_{\mathcal{P}_m^*}\{\mathcal{U}(z_m)\}$ be the expected gain of participant $m$ when active, $m \in [K]$. The necessary and sufficient condition for reaching a pricing consensus is the existence of a participant, say participant 1, that satisfies

$$\mu_1 - u_j \geq \mathbb{E}_{\mathcal{P}_1^*}\{C(z_1)\} + \mathbb{E}_{\mathcal{P}_j^*}\{(K - 1)C(z_j)\} \tag{72}$$

for all $j \neq 1$. In particular, assume the linearity $\mathcal{U}(z) \triangleq u \cdot z$ and $C(z) = c \cdot z$. Inequality (72) is equivalent to

$$c \leq \min_{j \neq 1} \frac{u \cdot (\mu_1 - \mu_j)}{\mu_1 + (K - 1)\mu_j}.$$

*Proof.* The necessary and sufficient conditions for participant 1 to be active with consensus are: 1) from the system's perspective, participant 1 can maximize the expected collaboration gain; 2) from participant 1 perspective, its expected profit serving as an active participant is better than that of serving as non-active participant; from other participants' perspectives, the expected earn of being active is no better than being active. Specifically, we have the following conditions:

Perspective of collaboration gain 1 :
$$\mathbb{E}_{\mathcal{P}_1^*}\{\mathcal{G}(z_1)\} \geq \mathbb{E}_{\mathcal{P}_j^*}\{\mathcal{G}(z_j)\}, \forall j \neq 1 \tag{73}$$

Perspective of Participant 1 :
$$\mathbb{E}_{\mathcal{P}_1^*}\{(K - 1)C(z_1) + \mathcal{G}(z_1)\} \geq \mathbb{E}_{\mathcal{P}_j^*}\{-C(z_j) + \mathcal{G}(z_j)\}, \forall j \neq 1 \tag{74}$$

Perspective of Participant $j \neq 1$ :
$$\mathbb{E}_{\mathcal{P}_1^*}\{-C(z_1) + \mathcal{G}(z_1)\} \geq \mathbb{E}_{\mathcal{P}_j^*}\{(K - 1)C(z_j) + \mathcal{G}(z_j)\}. \tag{75}$$

Here, Inequality (73) states that if there exists an active participant with consensus, it must achieve the maximal collaboration gain. Inequality (75) is equivalent to Inequality (72). By the assumption of non-negativeness of $C$, Inequality (75) implies Inequality (73), which further implies Inequality (74). This proves the first part of the theorem.

In particular, assuming $\mathcal{G}(z) = g \cdot z$ and $C(z) = c \cdot z$, Inequality (75) can be equivalently written as

$$-c\mu_1 + g\mu_1 \geq \max_{j \neq 1}\{(K - 1)c\mu_j + g\mu_j\}, \tag{76}$$

which can be equivalently written as

$$c \leq \min_{j \neq 1} \frac{g \cdot (\mu_1 - \mu_j)}{\mu_1 + (K - 1)\mu_j}. \tag{77}$$

This concludes the proof. □

