# OpenReview forum: "Incentivized Collaborative Learning: Architectural Design and Insights"
_ICLR.cc/2024/Conference — Submitted to ICLR 2024_

### Official Review · Reviewer_R699 · 2023-10-27

**Soundness:** 3 good
**Presentation:** 2 fair
**Contribution:** 2 fair
**Rating:** 5
**Confidence:** 4

**Summary:**

This paper proposes an architecture framework for collaborative learning with the aim of incentivizing collaboration among multiple entities while maximizing the utility of the coordinator. The framework is realized by formulating a pricing plan, which determines participants’ participation cost, and a selection plan, which selects active participants to determine the collaboration outcome. The conditions of Nash equilibriums and also the optimization objectives for the system are derived in the paper. The authors have also empirically shown the versatility of the framework by applying it to three concrete learning scenarios of interests, including federated learning, assisted learning and multi-armed bandits.

**Strengths:**

1. The motivation considering the incentives of both the coordinator/system and the participants is sound.
2. The framework includes important stages of the incentivized learning pipeline, such as pricing, selection and rewarding, which is rather comprehensive.
3. The paper has shown the application to three scenarios to incentivize FL, AL and MAB and presented empirical evidence for the effects of pricing and selection designs in the framework.

**Weaknesses:**

1. The novel contributions and the new insights from the unified framework need further clarification.
2. The pricing plan formulation and its dependency on the individual outcomes of active and non-active participants is not clearly elaborated in the paper.

The details for the weaknesses raised above are elaborated below in the Questions.

**Questions:**

1. This work proposes a grand framework with abstract terminologies that unifies existing formulations for the incentivization problem into a unified framework. However, what are the new insights that can be derived because of this unification and cannot be observed otherwise? I can see that the work still mainly relies on deriving Nash equilibriums with individual rationalities (IR) constraints, and strives to maximize some system utility (maybe more flexible with hyperparameters). These concepts are commonly seen in the existing literature that the authors have cited. And mutual benefits of the coordinator (or, system) and the participants (or, clients) are not rare in existing works, either. Therefore, I wonder what are the new insights? This point is important to assess the significance of this paper.
2. Could you elaborate on the specific meaning and implications of “prior works has often focused on designing an incentive as a separate problem based on an existing collaboration scheme, instead of treating incentive as part of the learning itself”? This is stated at the end of the “Related Works” section.
3. The pricing plan $\mathcal{P}$ looks at the realized collaboration gain from the outcomes of the active participants in $I_A$ to charge all participants in $I_P$. Does the pricing differentiate participants with high $z_m$ from those with low $z_m$?
4. It appears strange to me that the pricing for the non-active participants depends on the individual gains of the active participants $z_m$, stated in (1). Please correct me if I am mistaken.
5. Under the formulation of this paper, the incentive of participation for $m$ only depends on the utility income of $m$ himself minus the participation cost. This means that the client is incentivized as long as (9) is fulfilled. However, in practice, a candidate’s incentive should also depend on other candidates. For example, knowing that others have a higher expected gain with lower-quality data can deter participation. How does the framework address this scenario?
6. Clarification: At the end of Page 5, it is stated that “if not selected, it will become an inactive participant with zero gain, which will contribute to the system’s profit but not harm the collaboration.” Why would the gain be zero? All participants will receive $z_{I_A}$ and there might be a large utility income gain for this “inaccurate candidate”.
7. How does Theorem 1 translate to functional forms of the pricing plan in practice?
8. Why is it acceptable and reasonable to assume the utility functions (e.g., utility income $\mathcal{U}$) of the candidates are known to the system for pricing and selection? Also, how practical is it to calculate $E [ \mathcal{U}(z_{I_A}) - \mathcal{U}(z_m) ]$?
9. It was mentioned in Remark 5 that “if following the above-optimized selection plan, will not select it as active”. However, I could not find descriptions in the main text about the selection plans that make a decision based on the prices $\mathcal{P}_m$ or local gain $z_m$. I only see descriptions about randomized selection. Could you point me to the relevant sections or elaborate here?

[Minor]
1. Better not to overload the collaborative gain function $\mathcal{G}$ to take an individual $z_m$ as input, since $z_m$ should also depend on the outcomes of other active participants in $I_A$?
2. Some other notations are confusing. For example, $I_P = Incent_m (\mathcal{P})$, here $I_P$ is a set while $Incent_m()$ outputs whether a client $m$ is incentivized to participate.

---

> ### Author Response · Authors · 2023-11-23
> **Thank you for your constructive comments.**
>
> **1**. This work proposes a grand framework with abstract terminologies that unifies existing formulations for the incentivization problem into a unified framework. However, what are the new insights that can be derived because of this unification and cannot be observed otherwise? I can see that the work still mainly relies on deriving Nash equilibriums with individual rationalities (IR) constraints, and strives to maximize some system utility (maybe more flexible with hyperparameters). These concepts are commonly seen in the existing literature that the authors have cited. And mutual benefits of the coordinator (or, system) and the participants (or, clients) are not rare in existing works, either. Therefore, I wonder what are the new insights? This point is important to assess the significance of this paper.
>
> **Response**: Thanks for your query about new insights from our framework. We extract some concrete examples from the paper to demonstrate the insights of our approach.
>
> Insight 1: suitable incentives can improve collaboration performance. Unlike prior work that treats incentives as a separate problem, our framework integrates incentives into the learning process itself. For example, in the federated learning use case, we developed a theory-guided algorithm for simultaneously optimizing the pricing plan and performing federated learning (Appendix D). The insight is that a client, especially an adversarial client, will opt out of the selection process if participation costs are too high. This reduces system exploration costs and improves collaboration efficiency.
>
> In the assisted learning use case, we showed the insight that entities could be incentivized to autonomously collaborate to enhance personal learning objectives, even without a central coordinator (Section 3.2.1 and Appendix E).
>
> In the multi-armed bandit use case (Appendix F), we showed the insight that screening at both participation and selection stages could simultaneously achieve the following goals: 1) prevent the laggard arms from participating by leveraging their incentives, which can further improve the collaboration gain, 2) encourage the medium-performing arms to participate to enjoy the collaboration gain, which can boost the overall profit, and 3) reward the top-performing arms to participate to make sure that they can be consistently selected to be active. Consequently, incentives reduce the complexity of exploration, which can further improve the quality of generated outcomes and engage a broader range of entities.
>
> Insight 2: Each participant can simultaneously play the roles of contributor and beneficiary of the collaboration gain. Each entity’s participation cost can be positive or non-positive to reward those who contribute positively and charge those who hinder collaboration or disproportionately benefit from it. An interesting case is to have the system incur a net zero cost, as exemplified in Section 3.2.2. The insight is that a system can still engage entities to achieve the maximum collaborative gains even without incurring net costs.
>
> Insight 3. The joint architecture of pricing and selection plans could manage free-riders and adversarial participants.
>
> A free-rider is an entity that hopes to participate and enjoy the collaboration gain realized by other more capable participants with a relatively small participation cost. By ensuring a minimum probability of being active and incurring a related cost, one could effectively prevent entities from exploiting the system without contributing. Additionally, for adversarial participants, the pricing plan may impose a high cost whenever the realized local gain revealed after the collaboration exceeds a pre-specified threshold, so no adversary would dare to risk paying an excessively high cost after participating in the game.
>
> These insights, derived from our case studies and theoretical work, illustrate how the unified ICL framework fosters win-win situations in collaborative settings, a perspective not extensively explored in existing literature. We believe these contributions significantly advance the understanding and application of incentivization in collaborative learning.

---

> > ### Author Response · Authors · 2023-11-23
> >
> > **2**. Could you elaborate on the specific meaning and implications of “prior works has often focused on designing an incentive as a separate problem based on an existing collaboration scheme, instead of treating incentive as part of the learning itself”? This is stated at the end of the “Related Works” section.
> >
> > **Response**: Thank you for the opportunity to clarify this statement. We were referring to the incentive mechanisms that are developed independently of the learning algorithms, e.g., creating rewarding strategies to encourage participation, devising payment allocations based on contribution evaluation, or setting objectives to maximize the number of incentivized clients. These approaches generally view incentives as an adjunct to the primary learning process.
> >
> >
> > **3**.  The pricing plan looks at the realized collaboration gain from the outcomes of the active participants to charge all participants. Does the pricing differentiate participants with high $z_m$ from those with low $z_m$?
> >
> > **Response**: Thank you for this question. Yes, our pricing plan is designed to differentiate between active participants based on their contribution levels to ensure proper incentives. An example of this is Equation (18) of Appendix D. Here, participants with higher $z_m$ values, indicating greater contributions, can pay less costs or receive more rewards than those with lower values. For participants who are not active, the pricing is more uniform since their specific contributions (realized gains) are not directly observable. However, they are still incentivized to contribute, considering the potential of being selected as active and thus facing related costs. This aspect is further illustrated in the multi-armed bandit example in Appendix F, where the potential of incurring costs influences participant behavior even when they are not currently active.
> >
> >
> > **4**. It appears strange to me that the pricing for the non-active participants depends on the individual gains of the active participants $z_m$, stated in (1). Please correct me if I am mistaken.
> >
> > **Response**: Thank you for raising this important point. You are correct in your understanding that the pricing for non-active participants can be influenced by the individual gains of the active participants, as reflected in (1). This design choice is intentional and stems from a key insight of our framework. The rationale is that when active participants achieve high performance, contributing significantly to the collaboration gain, they are deemed to merit greater rewards. These rewards could be, in part, sourced from the non-active participants. Non-active participants, in anticipation of potentially becoming active in future rounds and benefiting from the collaboration gains, are generally more willing to contribute to this reward pool. This approach ensures that all participants, active or not, are invested in the success of the collaborative effort and that high performers are appropriately compensated for their significant contributions. We hope this explanation clarifies the pricing mechanism in our framework.
> >
> >
> > **5**. Under the formulation of this paper, the incentive of participation for $m$ only depends on the utility income of $m$ himself minus the participation cost. This means that the client is incentivized as long as (9) is fulfilled. However, in practice, a candidate’s incentive should also depend on other candidates. For example, knowing that others have a higher expected gain with lower-quality data can deter participation. How does the framework address this scenario?
> >
> > **Response**: Thank you for pointing out the aspect of interference between participants in our incentive mechanism. We recognize that this dynamic is not explored in our current work. Your insight is valuable, and we will mention this as future research.
> >
> >
> > **6**. Clarification: At the end of Page 5, it is stated that “if not selected, it will become an inactive participant with zero gain, which will contribute to the system’s profit but not harm the collaboration.” Why would the gain be zero? All participants will receive $z_{I_A}$ and there might be a large utility income gain for this “inaccurate candidate”.
> >
> > **Response**: Thanks for pointing out this. It was a typo and we will remove “with zero gain”.

---

> > > ### Author Response · Authors · 2023-11-23
> > >
> > > **7**. How does Theorem 1 translate to functional forms of the pricing plan in practice?
> > >
> > > **Response**: Thanks for your question. Theorem 1 can be translated to particular forms of the pricing plan through more explicit assumptions of how the collaboration gain can be calculated or approximated. The paper demonstrates some specific use cases. For example, in the federated learning use case, we used Theorem 1 to derive Theorem 2 under a large-participation assumption (so a law of large numbers could be applied for approximation). We further translated it into an algorithm to automatically identify a concrete pricing plan from a parametric form of the pricing plan (Appendix D.1). We also translated Theorem 1 in the multi-armed bandit use case to quantify the conditions for the pricing plan (Appendix F.2).
> > >
> > > **8**. Why is it acceptable and reasonable to assume the utility functions (e.g., utility income $U$) of the candidates are known to the system for pricing and selection? Also, how practical is it to calculate $E(U(z_{I_A}) - U(z_m))$?
> > >
> > > **Response**: Thank you for raising this point. It is true that in the current framework, we need the utility functions to be known for designing the pricing and selection in concrete use cases. This assumption helps in demonstrating the theoretical aspects of our model more clearly. However, I acknowledge that in real-world scenarios, the exact utility functions of participants may not always be fully known. In those cases, utility functions can be approximated or inferred through data analysis, historical behavior patterns, or market research. We see a valuable future direction of research in developing methods to dynamically learn these utility functions within our framework. As for the calculation of the expected utility difference, we would like to refer to the same response in the above comment 7.
> > >
> > >
> > > **9**. It was mentioned in Remark 5 that “if following the above-optimized selection plan, will not select it as active”. However, I could not find descriptions in the main text about the selection plans that make a decision based on the prices or local gain.
> > >
> > > **Response**: Thanks for pointing out this lack of clarity. We made a high-level statement that an entity reporting a low potential contribution would generally be less likely to be selected by the system under the optimized selection plan. This is based on the premise that the selection mechanism is designed to favor entities that are expected to contribute more significantly to the collaboration gain. We will revise this section to improve clarity.
> > >
> > > **10**. Better not to overload the collaborative gain function to take an individual $z_m$ as input, since $z_m$ should also depend on the outcomes of other active participants?
> > >
> > > **Response**: Thank you for your comment. We would like to clarify that in Section 2.2, the definition of collaboration gain function $G$ takes individual outcomes $x_m$ (rather than $z_m$) for $m \in I_A$ as input. This is designed to capture the interaction and interference between active participants. To maintain notational simplicity, we have also used the same symbol $G$ in a different context to represent the gain of an individual outcome. In this case, $G$ refers to the gain achieved when an individual operates independently, without being part of the collaborative effort.
> > >
> > > **11**. Some other notations are confusing. For example, $I_P = Incent_m(\mathcal{P})$.
> > >
> > > **Response**: Thank you for pointing out this issue. We have fixed the notation by replacing “Incent_m” with “Incent” and proofread the paper. We have conducted a thorough proofreading of the manuscript to ensure that all notations are used consistently.

---

### Official Review · Reviewer_cras · 2023-10-30

**Soundness:** 2 fair
**Presentation:** 1 poor
**Contribution:** 2 fair
**Rating:** 5
**Confidence:** 4

**Summary:**

Starting from the significant challenge of collaborative learning, this paper aims to solve how to effectively motivate multiple entities to collaborate before any collaboration occurs. And proposed ICL framework. This work elaborates on the roles, processes, and principles of the games used in the framework, and proved the effectiveness of ICL through mathematical derivation. By using different pricing or selection plans in the experiments, the authors discussed how incentive settings affect the effectiveness of the framework.

**Strengths:**

1.	The authors proposed a clear incentivized collaborative learning framework, along with detailed descriptions of the roles and principles of ICL.
2.	The authors gave sufficient and detailed mathematical derivation to prove the effectiveness of ICL.
3.	Many experiments were conducted to validate the effectiveness of the proposed ICL framework and analyse the influences from pricing and selection plans.

**Weaknesses:**

1. The discussion in the paper cannot fully reflect the superiority of the proposed framework compared to previous methods. For example, there seems no specific comparative experiment to confirm that the proposed ICL framework is more efficient than previous works.
2. It’s kind of confusing that the description of the experiements is less detailed about their design. For example, the employed model and the meaning of the metrics in the first experiment are not very clear.
3. Experiment settings are kind of insufficient to support all the contributions, such as the discussion about the influences from the selection plans, while most of the selection plan is based on Bernoulli distribution. The discussion about how to select appropriate pricing and selection plans is also insufficient.

Overall, the research problem is meaningful and well-defined. However, the paper is somewhat lack of clarity to make readers understand the superiority of the proposed method fully and easily.

**Questions:**

1.	Can you give more explicit evidences that the proposed ICL framework utilizes the incentive mechanism more effectively than previous works?
2.	I wish the experiment settings can be more detailed written in the main body of the paper.

---

> ### Author Response · Authors · 2023-11-23
> **Thank you for your constructive comments.**
>
> **Comment 1**: The discussion in the paper cannot fully reflect the superiority of the proposed framework compared to previous methods. For example, there seems to be no specific comparative experiment to confirm that the proposed ICL framework is more efficient than previous works.
>
>
> **Response**: Thank you for your valuable feedback regarding the comparative analysis of our ICL framework. We have included several experimental studies that compare incentivized and nonincentived/standard collaborative learning in federated learning (FL), assisted learning (AL), and multi-armed bandit (MAB) settings. We did not compare it with other incentivation techniques because the introduced architectural framework fundamentally differs from previous methods. The ICL framework is designed to address the under-explored area of when and why incentive mechanisms can enhance collaboration performance. The core contribution of our work is the establishment of general principles and insights into incentivized collaboration. Our experimental validation is strategically designed to corroborate the theoretical insights and principles developed in the ICL framework. Our experiments are not merely about efficiency comparisons but are aimed at demonstrating the practical applicability and benefits of the proposed framework in a variety of settings.
>
>
> **Comment 2**: It’s kind of confusing that the description of the experiments is less detailed about their design. For example, the employed model and the meaning of the metrics in the first experiment are not very clear.
>
> **Response**: Thank you for pointing this out. We will add more details such as model architectures, learning rates, and optimizers in the revised appendix. In particular, in the first experiment shown in Figure 2, we visualized the top-1 accuracy against communication rounds on the CIFAR10 dataset to show that a suitably incentivized FL algorithm in our framework can converge faster and better than the non-incentivized counterpart.
>
>
> **Comment 3**: Experiment settings are kind of insufficient to support all the contributions, such as the discussion about the influences from the selection plans, while most of the selection plan is based on Bernoulli distribution. The discussion about how to select appropriate pricing and selection plans is also insufficient.
>
> **Response**: Thank you for your insightful feedback. In our study, we focused on a probabilistic selection plan as detailed in Sections 2.4.2 and 3.1.1. This approach was chosen due to its broad applicability in various collaborative learning scenarios. In Remark 5, we provided a discussion of how these plans can be effectively utilized to mitigate issues like free-riding and adversarial behavior. However, we acknowledge that our analysis does not extend to more complex, jointly dependent selection probabilities, which presents a challenging yet exciting avenue for future research. Regarding the pricing plans, we explored specific functional forms in our three use cases. These plans were chosen to align with the unique requirements of each scenario, highlighting the framework's adaptability. For example, in the federate learning experiments in Appendix D, we postulated a parametric form of the pricing plan in Equation (18) and proposed to optimize it online under the ICL framework. Such optimization will automatically identify/select a concrete pricing plan to operate for the next round of collaboration. Again, we appreciate your feedback which has improved the clarity of the work. If you have any suggestions or specific areas you believe would benefit from further exploration, we would greatly appreciate your input.

---

### Official Review · Reviewer_npuS · 2023-11-01

**Soundness:** 2 fair
**Presentation:** 2 fair
**Contribution:** 2 fair
**Rating:** 3
**Confidence:** 4

**Summary:**

This work investigates incentivized collaborative learning where there are candidates that are potentially looking into joining the federation, the participants who will get the reward from the actual outcomes of the collaboration, and the active participants who participate in the training. The work defines a coordinator who orchestrates the participation of the clients and the pricing plan and profit, and depending on these components, propose to maximize the system-level profit under constraints of individual clients' incentives. The work investigates different use cases of the proposed incentivized collaborative learning framework along with analysis on robustness and accuracy.

**Strengths:**

- The work investigates an interesting area in collaborative learning regarding client incentives and monetary compensations and cost analysis. The work proposes a framework for incentivized collaborative learning where federated learning, assisted learning, and MAB all come under their umbrella.

- The work provides theoretical results, although limited to specific scenarios such as the three-entity setting.

- The work evaluates their framework under robustness against byzantine attacks and scenarios where there are both competing and non-competing clients.

**Weaknesses:**

- A major concern I have over the work is that in stage 1 of the method, the coordinator needs to set a pricing plan based on prior knowledge of candidates potential gains from previous rounds. This means that first the client needs to participate first to know its incentives, and moreover, if the potential gain is erroneous, the ICL framework may not be able to properly incentivize the clients. This becomes even more trickier when clients have the flexibility to opt-in or opt-out which can often be the case for incentivized collaborative learning settings.

- Another concern I have is that the work did not compare their method against other relevant work for incentivization in collaborative learning such as
 [1] Yae Jee Cho, Divyansh Jhunjhunwala, Tian Li, Virginia Smith, and Gauri Joshi. To federate or not to federate: Incentivizing client participation in federated learning. arXiv preprint arXiv:2205.14840, 2022.
[2] Avrim Blum, Nika Haghtalab, Richard Lanas Phillips, and Han Shao. One for one, or all for all: Equilibria and optimality of collaboration in federated learning. In International Conference on Machine Learning, pp. 1005–1014. PMLR, 2021.
[3] Rachael Hwee Ling Sim, Yehong Zhang, Mun Choon Chan, and Bryan Kian Hsiang Low. Collaborative machine learning with incentive aware model rewards. In International Conference on Machine Learning, pp. 8927–8936. PMLR, 2020.
Especially for parts of the work such as 3.1.1 which aims for large participation approximation works like [1] seem relevant and other parts such as section 3.2, [2,3] seems relevant. It seems strange to me that the work does not compare their work with such relevant line of work.

- Lastly, the work seems mainly theoretical since the experimental validation is rather limited. However, the assumptions they use for the theoretical work such as having a three entity setting is rather restrictive. Moreover the implications of the main theoretical results such as Theorem 2 and 4 is unclear to me. In what conditions it is guaranteed that the clients benefit from the system for each corresponding theorems?

Due to these main reasons I am leaning towards rejection. I look forward to the author's rebuttal phase and discussion.

**Questions:**

See weaknesses above.

---

> ### Author Response · Authors · 2023-11-23
> **Thank you for the constructive comments.**
>
> **Comment 1**: A major concern I have over the work is that in stage 1 of the method, the coordinator needs to set a pricing plan based on prior knowledge of candidates potential gains from previous rounds. This means that first the client needs to participate first to know its incentives, and moreover, if the potential gain is erroneous, the ICL framework may not be able to properly incentivize the clients. This becomes even more trickier when clients have the flexibility to opt-in or opt-out which can often be the case for incentivized collaborative learning settings.
>
> **Response**: Thank you for this comment. It is indeed a valid concern that the coordinator needs to set the pricing plan according to prior knowledge of candidates’ potential gains in order to optimize the system gain. In the experimental studies, we implemented this for each round using realized gains from previous rounds. Nevertheless, this does not necessarily mean that a client needs to participate first to know its incentives. For example, in the multi-armed bandit (MAB) use case, each candidate decides its incentive by observing the gains achieved by selected arms and contrast with its local potential to generate rewards; if it does not know its potential to generate reward, it could locally simulate to obtain an estimate of its potential. Similarly, in FL, a client could choose to run a local update to decide its incentive to participate, should local computation cost be not a matter of concern. In the paper, we did not specify the concrete way of obtaining such prior knowledge in the framework description, and instead, mentioned specific possible choices when introducing the three use cases. We will incorporate your comment in the revision.
>
> **Comment 2**: Another concern I have is that the work did not compare their method against other relevant work for incentivization in collaborative learning such as [1] Yae Jee Cho, Divyansh Jhunjhunwala, Tian Li, Virginia Smith, and Gauri Joshi. To federate or not to federate: Incentivizing client participation in federated learning. arXiv preprint arXiv:2205.14840, 2022. [2] Avrim Blum, Nika Haghtalab, Richard Lanas Phillips, and Han Shao. One for one, or all for all: Equilibria and optimality of collaboration in federated learning. In the International Conference on Machine Learning, pp. 1005–1014. PMLR, 2021. [3] Rachael Hwee Ling Sim, Yehong Zhang, Mun Choon Chan, and Bryan Kian Hsiang Low. Collaborative machine learning with incentive aware model rewards. In the International Conference on Machine Learning, pp. 8927–8936. PMLR, 2020. Especially for parts of the work such as 3.1.1 which aims for large participation approximation works like [1] seem relevant and other parts such as section 3.2, [2,3] seem relevant. It seems strange to me that the work does not compare their work with such a relevant line of work.
>
> **Response**: Thank you for highlighting these excellent papers. While they are relevant as they all study incentives in federated learning from different perspectives, there are some essential differences with our work. Specifically, the main objective in [1] is to increase the number of incentivized clients by maximizing a proxy of the Incentivized Participation Rate. The section 3.1.1 in our work established a large-sample analysis of the equilibrium condition and an individual entity’s incentive under that. From the version of [1] we read, we could not find a large sample analysis similar to our section 3.1.1. Reference [2] studies the (centralized) resource allocation problem that aims to minimize the total social resource while meeting individual constraints. This is different from our section 3.2 that studies a decentralized incentive setting regarding how entities could be incentivized to work together autonomously. Reference [3] proposes to reward a federated learning client based on Shapley value and provide a model as a reward. We have cited the above works and mentioned their relevance and difference in the paper. Thank you again for pointing out these references.

---

> > ### Author Response · Authors · 2023-11-23
> >
> > **Comment 3**: Lastly, the work seems mainly theoretical since the experimental validation is rather limited. However, the assumptions they use for theoretical work such as having a three entity setting is rather restrictive. Moreover the implications of the main theoretical results such as Theorem 2 and 4 is unclear to me. In what conditions is it guaranteed that the clients benefit from the system for each corresponding theorems?
> >
> > **Response**: Thank you for your comments. Section 3.1.1 “Consensus of Competing Candidates” studied the 3-entity setting only for developing the insights of how decentralized collaboration could be attained under suitable incentives. The results can be extended to multiple entities in a straightforward way. We would like to comment more on the implications of Theorems 2 and 4. Theorem 2 is a result to approximate the equilibrium conditions in Section 2.4 under a large number of participating clients. It is a very important result because it quantities the system's marginal profit increase attributed to an entity. In Appendix D, we had elaborated on this idea and proposed a concrete incentivized federated learning algorithm built on Theorem 2, whose  empirical performance was also studied. Theorem 4 develops a necessary and sufficient condition to reach a consensus among the participants. Specifically, the condition is the existence of a participant such that when it is active, the collaboration gain is maximized and every participant sees that the participation cost is worthwhile (to gain individual profits).
> >
> > We appreciate your insightful feedback and hope that the above explanations have clarified our approach and findings. We are open to further discussions and would be glad to explore any additional questions or ideas you might have.

---

### Meta-Review · Area_Chair_Qjhm · 2023-12-05

**Metareview:**

This paper proposes an incentivized collaborative learning framework that can account for participants' costs, selection, and rewards, and applies it to federated learning, assisted learning, and multi-armed bandit.


**STRENGTHS**

(1) This paper addresses an important and interesting area of incentives in collaborative machine learning.

(2) This paper proposes a comprehensive framework that accounts for participants' costs, selection, and rewards.

(3) The proposed framework can be applied to several problems including federated learning, assisted learning, and multi-armed bandit.


**WEAKNESSES**

The reviewers have all agreed that the major concern of this work is the considerable lack of clarity, which requires a major revision of this paper. Though the authors have attempted to clarify the questions in their rebuttal, the clarifications require more concrete details and greater elaboration. For example, the insights provided in the rebuttal should be described in a way that connects easily with that of machine learning concepts and practice. More details and recommendations are given below.

(1) In particular, there is a lack of detailed qualitative and quantitative comparisons with related work. How would the proposed framework compare with or differ in the general principles, insights, and problem settings from the others that also develop incentives in collaborative learning? A detailed discussion on the pros and cons would be necessary here, especially when the authors claim that they cannot be empirically compared.

(2) The clarity can be improved further if the authors provide more explanations on their pricing plan and its dependency on the individual outcomes of active and non-active participants.

(3) There is also a lack of details of the experimental settings as well as how they can support the listed contributions of this work.

(4) The authors need to also discuss the limitations of their work, such as that highlighted by a reviewer: "knowing that others have a higher expected gain with lower-quality data can deter participation" cannot be accounted for. The "utility functions of the candidates are known to the system for pricing and selection".

(5) Since the coordinator needs to set a pricing plan based on prior knowledge of candidates potential gains from previous rounds in order to optimize the system gain, can the authors expound on the implications of such a dynamic pricing plan formally?


The authors are encouraged to revise their paper to improve its clarity by considering the reviewers' feedback.

**Justification For Why Not Higher Score:**

The major concern of this work is the considerable lack of clarity, which requires a major revision of this paper.

**Justification For Why Not Lower Score:**

N/A.

---

### Decision · Program_Chairs · 2024-01-16

Reject